# Quantifying Context Bias in Domain Adaptation for Object Detection

**Hojun Son** *hojunson@umich.edu*
*University of Michigan Transportation Research Institute*
*University of Michigan*

**Asma Almutairi** *asmaalm@umich.edu*
*University of Michigan Transportation Research Institute*
*University of Michigan*

**Arpan Kusari** *kusari@umich.edu*
*University of Michigan Transportation Research Institute*
*University of Michigan*

**Reviewed on OpenReview:** https://openreview.net/forum?id=YRUOAOnraG

## Abstract

Domain adaptation for object detection (DAOD) has become essential to counter performance degradation caused by distribution shifts between training and deployment domains. However, a critical factor influencing DAOD—context bias resulting from learned foreground-background (FG–BG) association—remains underexplored. In this work, we present the first comprehensive empirical and causal analysis specifically targeting context bias in DAOD. We address three key questions regarding FG–BG association in object detection: (a) whether FG–BG association is encoded during training, (b) whether there is a causal relationship between FG–BG association and detection performance, and (c) whether FG–BG association affects DAOD. To examine how models capture FG–BG association, we analyze class-wise and feature-wise performance degradation using background masking and feature perturbation, measured via change in accuracy (defined as drop rate). To explore the causal role of FG–BG association, we apply do-calculus to FG–BG pairs guided by class activation mapping (CAM). To quantify the causal influence of FG–BG association across domains, we propose a novel metric—Domain Association Gradient—defined as the ratio of drop rate to maximum mean discrepancy (MMD). Through systematic experiments involving background masking, feature-level perturbations, and CAM, we reveal that convolution-based object detection models encode FG–BG association. The association substantially impacts detection performance, particularly under domain shifts where background information significantly diverges. Our results demonstrate that context bias not only exists but also causally undermines the generalization capabilities of object detection models across domains. Furthermore, we validate these findings across multiple models and datasets, including state-of-the-art architectures such as ALDI++. This study highlights the necessity of addressing context bias explicitly in DAOD frameworks, providing insights that pave the way for developing more robust and generalizable object detection systems.

## 1 Introduction

Domain adaptation for object detection (DAOD) has been studied extensively to enable object detectors to perform well on datasets with distribution shifts from the training data (Kay et al., 2024; Chen et al., 2022; Deng et al., 2021; Hoyer et al., 2023; Li et al., 2022b; Koh et al., 2021; Kalluri et al., 2023). It is widely recognized that there is an entanglement between background and foreground features in object

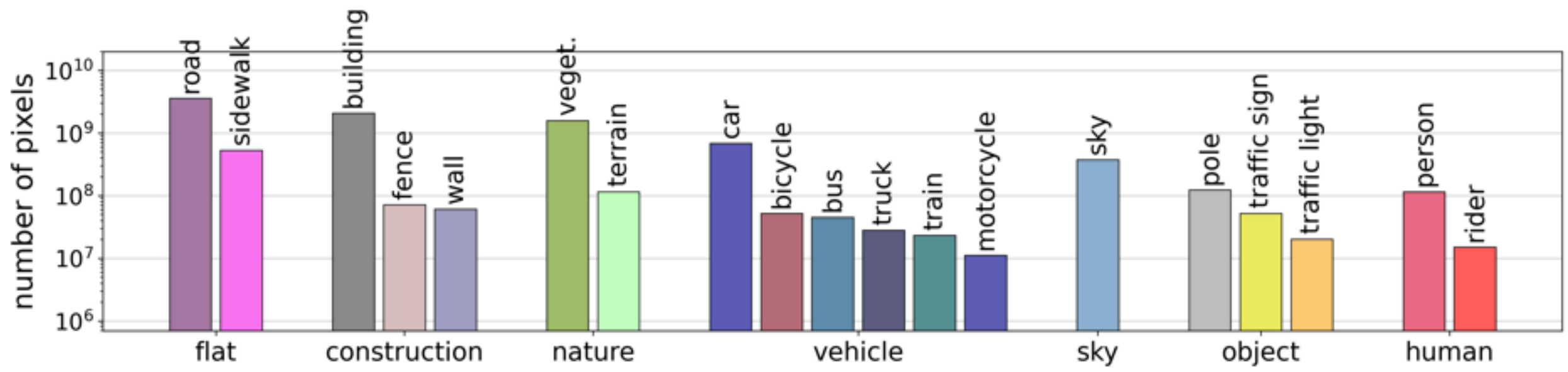

Figure 1: Background pixels constitute the largest proportion among all classes in the Cityscapes dataset (Cordts et al., 2016). Image is reproduced from the Cityscapes publication.

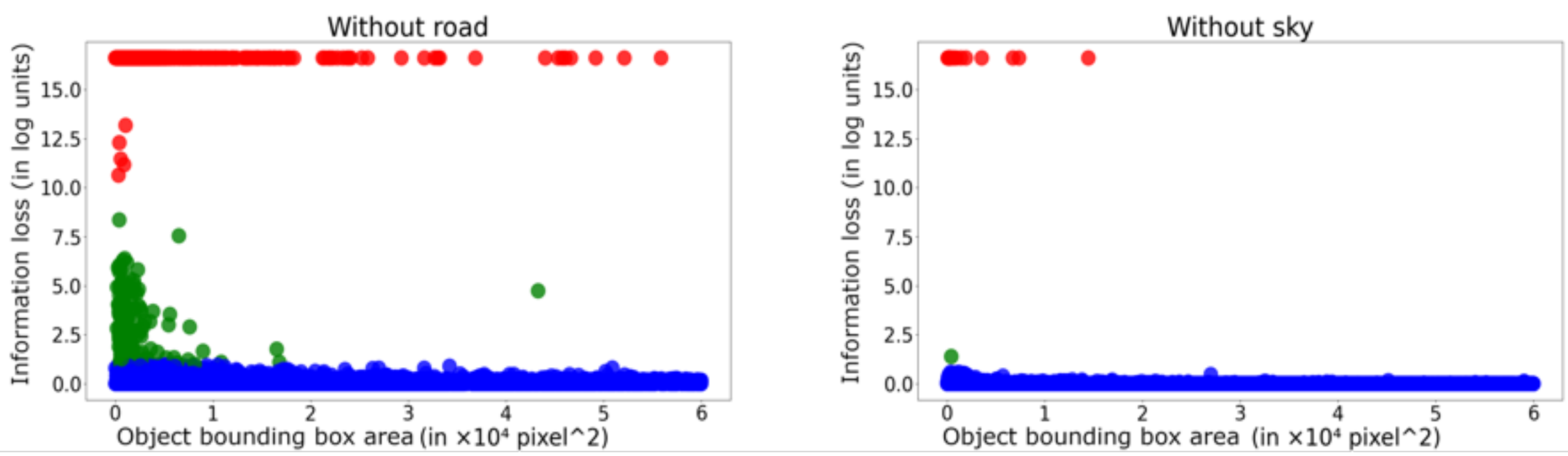

Figure 2: **Loss of information as a function of the bounding box area of the object.** Left: suppression of "road"; Right: suppression of "sky". The points are grouped into three clusters: red indicates missed detections (maximum information loss), green indicates partial matches (significant loss) and blue indicates no change. The label of "road" exhibits closer association with "car" than "sky".

detection, leading to a phenomenon called context bias in DAOD (Torralba & Efros, 2011; Divvala et al., 2009; Khosla et al., 2012; Zhang et al., 2024; Choi et al., 2012; Shetty et al., 2019). Here, significant differences in background features between the source and target domains can cause a notable decline in the quality and number of detections, even when the foreground features remain unchanged. Recent studies in image classification (Li et al., 2023; Aniraj et al., 2023) and segmentation (Zhu et al., 2024; Chen et al., 2021; Dreyer et al., 2023) have attempted to mitigate context bias by minimizing foreground-background (FG-BG) association. Oliva & Torralba (2007) demonstrated that context bias could result in the corruption of foreground objects by contextually correlated backgrounds, substantially degrading detection quality. However, no prior work has specifically analyzed the impact of context bias in DAOD. This work aims to address this gap.

In the realm of human cognition, the brain can accurately and instantly recognize FG–BG association without extensive training (Papale et al., 2018). Several studies, including Zhang et al. (2023); Poort et al. (2016); Papale et al. (2018); Huang et al. (2020) have investigated the processes of background suppression and foreground representation to understand the scene and temporal dynamics of foreground and background modulation in the brain. These insights can be applied to the field of computer vision for DAOD through a comprehensive analysis of how FG-BG associations are represented.

### 1.1 Our observations on context bias

To motivate our problem, we first look at the proportion of background features in autonomous driving datasets. As an example, for the Cityscapes dataset (Cordts et al., 2016), the number of pixels associated with built-up features (such as "road" and "sidewalk") is much greater than the number associated with foreground objects (see Fig. 1). Based on semantic segmentation outcomes (Alonso et al., 2021; Wang et al.,

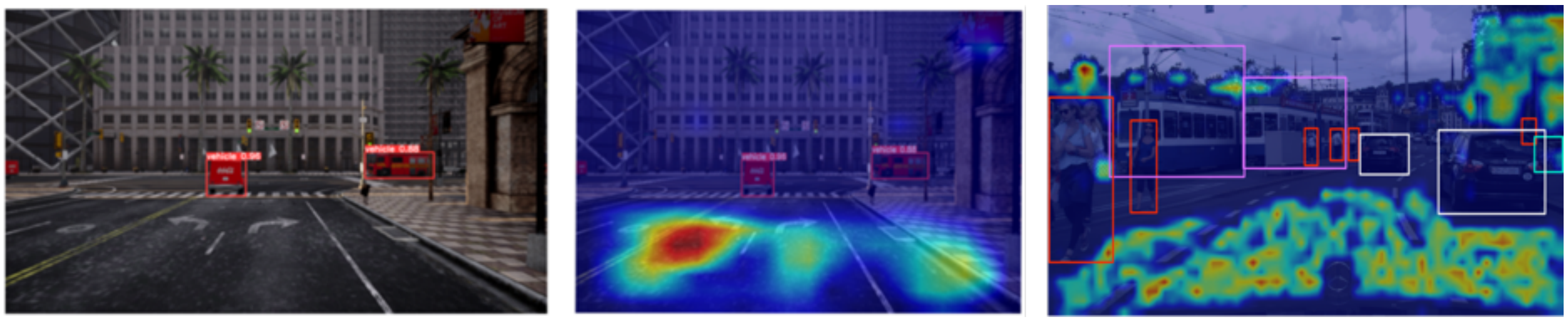

Figure 3: Left: 2D inference of the CARLA dataset using the YOLOv4 model, Center: CAM attention map of the inference using EigenCAM (Muhammad & Yeasin, 2020), Right: EigenCAM result of YOLOv11 trained on Cityscapes. Road and vegetation are significantly enhanced.

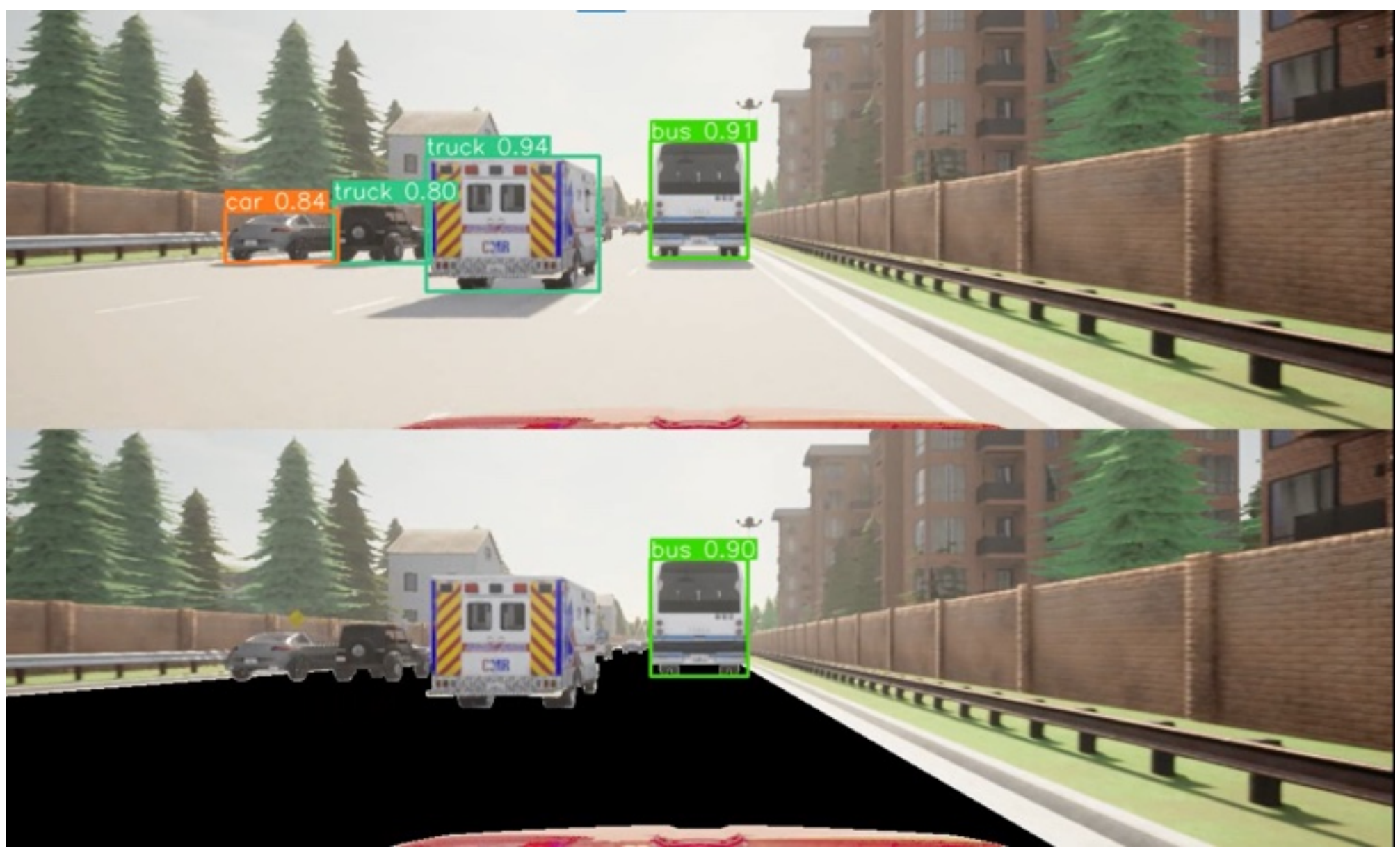

Figure 4: Masking the road on CARLA image and generating inference using YOLOv8 model. Top: 3 out of 4 vehicles are correctly detected. Bottom: road-masked image, where only one vehicle is detected.

2020; Erisen, 2024), "road" has the highest accuracy and lowest variability. As a motivating experiment, we aim to quantify the change in performance as a function of background masking for a real dataset. We use the second layer (`res2.2`) of the ResNet-50 backbone in Detectron2 (Wu et al., 2019), trained on the Cityscapes dataset for object detection. We hypothesize that `res2.2` effectively balances low- and high-level features for FG–BG association. The loss of information is computed when activated features for specific background regions are set to zero, using semantic labels and the ground-truth bounding-box area of the foreground objects. We define the performance drop, $\Delta\text{IoU}$, as $0 \leq \Delta\text{IoU} \leq 1$, and quantify the loss of information as the negative log of the complement of $\Delta\text{IoU}$, i.e., $-\log(1 - \Delta\text{IoU})$, which measures change in intersection over union with background masking. Figure 2 shows the performance drop associated with the removal of "road" as opposed to "sky". We find that the loss of information is much higher when "road" is suppressed compared to "sky", which means that "road" has more contextual association with "car", particularly when the "car" size is small.

We train a YOLOv4 detection model (Bochkovskiy et al., 2020) on a sample CARLA (Dosovitskiy et al., 2017) dataset collected under sunny conditions and provide inference on a separate CARLA dataset collected under cloudy conditions. We find, using class activation mapping (CAM), that the model focuses on the road in front of the vehicles rather than the vehicles themselves (see Fig. 3). Additionally, to determine whether this issue arises across different types of models, we conduct an analogous experiment in which we mask

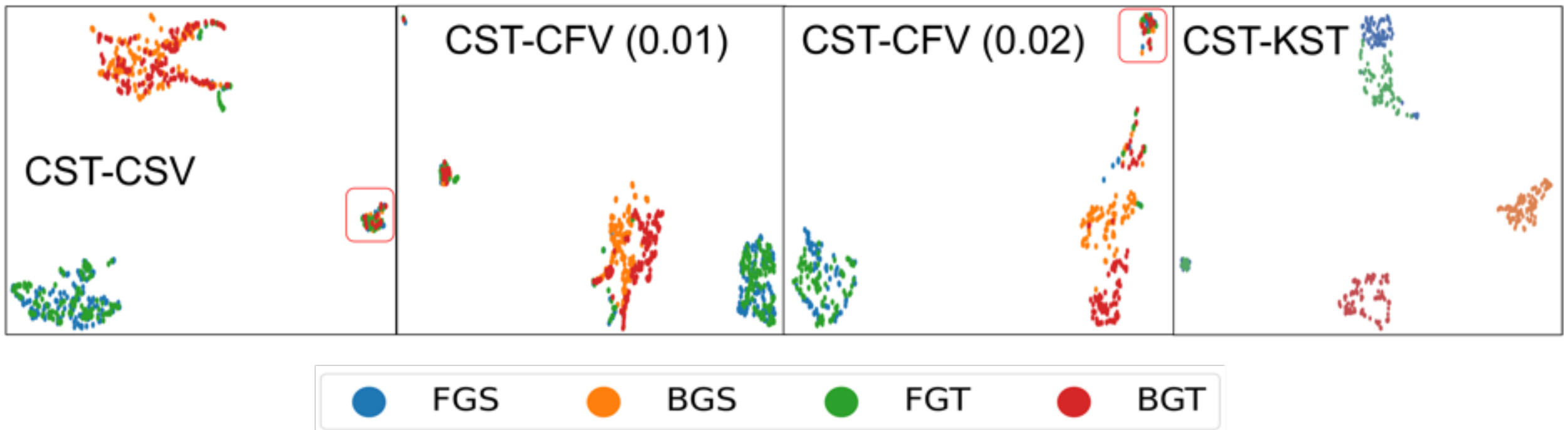

Figure 5: **UMAP feature embedding results.** "CST" stands for Cityscapes train, "CSV" for Cityscapes validation, and "CFV (0.0X)" for Foggy Cityscapes validation with different parameters for fogginess. "KST" means KITTI semantic train. We used the "car" label, as it is the most common category across the datasets. "FGS" is short for foreground in the source domain, and "BGS" is background in the source domain. "FGT" and "BGT" represent foreground and background in the target domain. Similar colors being intertwined indicate similar feature distributions.

the road pixels and find that the YOLOv8 model is unable to detect most vehicles that would otherwise be detected in the normal image (see Fig. 4). These results suggest that convolution-based neural network models may implicitly learn to associate vehicles with road environments, leading to poor performance in detecting vehicles when a different background is present.

In order to understand the spread of feature patterns across domains, we plot the foreground and background feature distributions of different domains using UMAP (McInnes et al., 2018). Figure 5 visualizes the foreground and background features from different domains in two dimensions. We use the features of "car" extracted at the $5^{th}$ ResNet layer (`res.5.2`) from various data distributions, assuming that `res.5.2` captures high-level foreground and background features.

Interestingly, background alignment differs across comparisons. As the target domain shifts away from the source domain, the background features become more separable than the foreground features. For the Cityscapes training (CST) and validation (CSV) datasets, foreground and background features are distinguishable but intermingled between the source and target domains. In the CST-CFV comparison, foreground features remain clustered together while background features are separable but overlapping. In the extreme case of CST-KST, foreground features from CST and KST are adjacent but non-overlapping, while background features are very distant from each other. The feature extraction process is illustrated in Section 3.

Prior studies (Choi et al., 2012; Torralba, 2003) have researched context bias for object detection and classification. The studies have pointed out that relying only on local features (foreground features in our case) has limitations, including degraded quality due to noise and ambiguity in the target search space. They extend the likelihood to incorporate context information surrounding the foreground, thereby enhancing object classification and detection by providing a stronger conditional probability, as in Equation 1. The conditional probability of the object ($O$) given the features ($f$) is given as:

$$P(O|f) = P(O|F, B) = \frac{P(F|O, B)P(O|B)}{P(F|B)} \tag{1}$$

where $F$ and $B$ denote the foreground and background features, respectively. However, it does not address DAOD issues like sim-to-real transfer, and the root causes of FG–BG association during training and inference remain unclear, especially given the causal relationships imposed post-detection.

The challenge with using a convolutional neural network (CNN) under the assumption of an identical and independent distribution (i.i.d.) to estimate likelihood is the inability to explicitly teach the model to learn each factor in a specific order (Schölkopf et al., 2021; Agrawal et al., 2019). In other words, the parameters

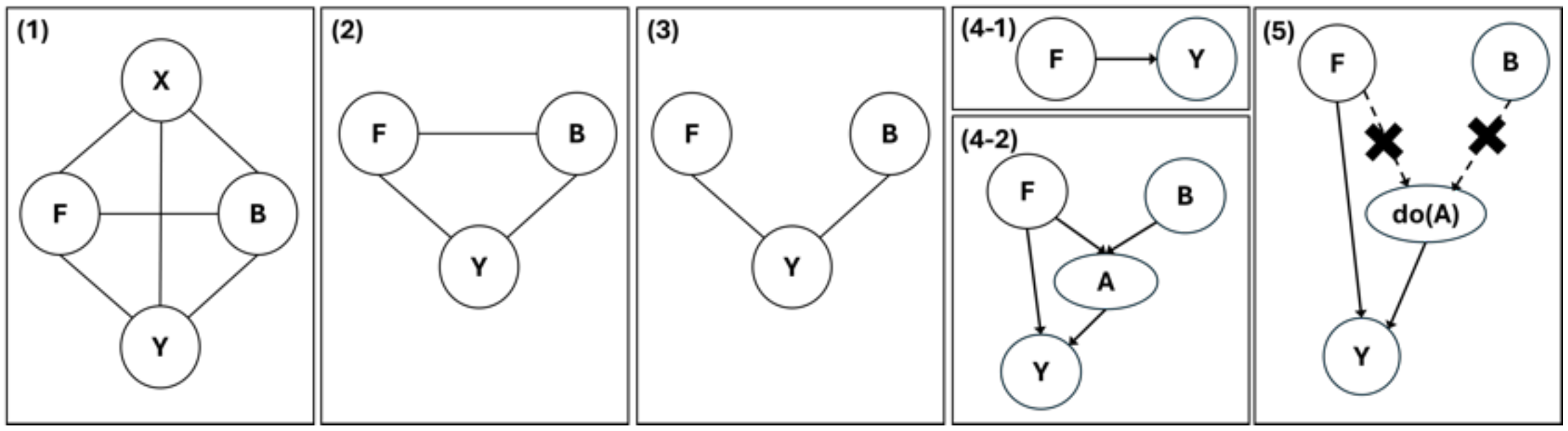

Figure 6: **Identification process to generate a causal graph in five steps.** (1): Initial phase, originating from a complete undirected graph, where “X” represents an image, “F” the foreground, “B” the background, and “Y” the outcome. (2): Result after removing “X” from further analysis, as the object detection model will process this input. (3): Step induced to illustrate the independence of “F” and “B”. (4-1): Optimal factor graph establishes that the outcome is exclusively related to “F”. (4-2) In contrast, preliminary experimental evidence (see Introduction) suggests the presence of FG-BG association, denoted by the variable “A”. (5): Factor graph incorporating do-calculus to quantify the causal effect of variable “A” on “Y”. By implementing the backdoor adjustment method, we can effectively measure the causal effect.

of a CNN can differ depending on how it is trained, as in equation 2. The modeling can also be interpreted as:

$$P(O|f) = P(O|B,F) = \frac{P(B|O,F)P(O|F)}{P(B|F)} \tag{2}$$

In CNNs, likelihood estimation is the process of finding the mean of a distribution given proper priors, which requires more samples to accurately estimate the true mean. This aligns with the principle that a more extensive and refined dataset, achieved through data augmentation, is crucial for better performance (Taylor & Nitschke, 2018). However, such datasets typically do not account for FG–BG association, which is subtle and difficult to capture during data collection. In summary, FG–BG association can disrupt the trained estimation process for each probability estimate, leading to performance degradation in target domains due to these broken associations.

From these observations and hypotheses, our fundamental questions are as follows:

### Q1. Is FG–BG association being inadvertently learned during the training process?

Deep learning identifies latent patterns that optimize objective functions, typically by maximizing data likelihood. During the feature extraction process, models may learn spurious or unexpected features if doing so improves predictive performance, without any understanding of causality (Bishop, 2006; Goodfellow et al., 2016; MacKay, 2003; Murphy, 2012). This underscores the importance of incorporating causal reasoning into deep learning frameworks to improve robustness and generalization (Schölkopf et al., 2021). From our motivation (see Fig. 2, 3, and 4), we address Q1 through two experiments by performing class-wise and feature-wise background removal.

These experiments are designed to capture the existence of FG–BG association under controlled conditions. Based on these findings, we ask: **To what extent does FG–BG association affect model accuracy?** To address this, we conduct a series of experiments aimed at capturing, representing, and quantifying the impact of FG–BG association across domains, which leads to our next set of questions (Q2 and Q3).

### Q2. Is there a causal relationship between FG–BG association and object detection?

We employ graph-based causal analysis to investigate the causal effect of FG–BG association on object detection performance. As illustrated in Figure 6, we construct a causal model using the PC algorithm (Glymour et al., 2019) to infer causal relationships within the object detection pipeline. In the context of a graphical causal model (F →Y ←B), the joint distribution $P(Y, F, B)$, can be factorized as either $P(Y|F, B)P(F|B)P(B)$ or $P(Y|F, B)P(B|F)P(F)$ (see panel (3) in Figure 6 and Equations 1, and 2). This represents a spurious factor that leads to FG–BG association in CNN training. Causal identification is further enabled via intervention analysis using do-calculus. The causal effect of FG–BG association on object detection performance can be expressed as $P(Y|\text{do}(A)) = \sum_F P(Y|A, F)P(F)$, following the backdoor adjustment formula. This effect can be quantified by the expected difference $\mathbb{E}[Y|A = 0, F] - \mathbb{E}[Y|A = 1, F]$, which captures the impact of FG–BG association ($A$) on detection outcomes ($Y$), conditioned on contextual features ($F$) (see Fig. 6). To this end, we design an experiment using interventions via CAM and instance masks. By combining CAM with ground-truth instance masks, we perform do-calculus interventions with backdoor adjustment to model the causal influence of FG–BG association on object detection accuracy. This approach allows us to control activated background regions for each instance using different thresholds.

### Q3. What is the impact of FG–BG association on DAOD and how can the effect be quantified?

While the causal association between FG-BG association and detection outcomes ($Y$) learned in source domains typically remains stable, the conditional distribution $P(Y|A)$ may shift due to changes in background distribution. This can weaken the causal strength in target domains, resulting in performance degradation. We quantify this effect using the Domain Association Gradient (Gradient, hereafter), which captures the impact of FG–BG association by combining intertwined features within and across domains. We note that previous domain adaptation research emphasizes foreground feature alignment exclusively (e.g., domain discriminator (Zhou et al., 2023) and moment matching (Peng et al., 2019)), and thus our effort is to highlight the overlooked role of FG-BG association.

In the causal graph shown in Fig. 6, FG-BG association is denoted by $A$ and detection performance by $Y$. The causal effect of $A$ on $Y$, analogous to the Average Treatment Effect (ATE) (Naimi & Whitcomb, 2023), can be given by the interventional contrast:

$$ATE = \mathbb{E}\big[Y \mid do(A = 1),\ F\big] - \mathbb{E}\big[Y \mid do(A = 0),\ F\big] \tag{3}$$

where $A = 1$ corresponds to the presence of the observed FG–BG association and $A = 0$ denotes the intervention that destroys this association. In practice, we estimate this effect using the drop rate ($\Delta Y$), defined as the decrease in detection accuracy when the background is intervened upon to disrupt FG–BG association while keeping the foreground object unchanged. Therefore, the drop rate can be seen as an empirical approximation to the average causal effect of $A$ on $Y$.

While the drop rate shows how much performance shifts under an intervention on $A$, it does not quantify changes in association. To disentangle these two factors, we use a Maximum Mean Discrepancy (MMD)-based term to measure the magnitude of the feature distribution shift between associated and non-associated features across inter- and cross-domain contexts. Specifically, the combination $\Delta MMD \approx MMD(\texttt{f2b}) + MMD(\texttt{b2f}) - MMD(\texttt{f2f}) - MMD(\texttt{b2b})$ captures the divergence between foreground ($f$) and background ($b$) features when the association is broken, compared to within-context variation (`f2f` and `b2b`). In other words, cross-context feature comparisons, such as `f2b` and `b2f`, are expected to yield higher MMD values than within-context comparisons, such as `f2f` and `b2b`, because the latter reflect similar contextual structures. However, if FG–BG association significantly influences object detection, the MMD of cross-context features is smaller than when the association is absent, yet it remains larger than within-context comparisons. A small $\Delta MMD$ value indicates a strong FG–BG association, whereas a large value implies a weak association. The Domain Association Gradient is then defined by Equation 4.

$$\text{Gradient} \approx \frac{\text{change in outcome } (\Delta Y)}{\text{association magnitude } (\Delta MMD)}. \tag{4}$$

The numerator measures the interventional change in the outcome, while the denominator, $\Delta MMD$, quantifies the magnitude of FG–BG association. Intuitively, a strong FG–BG association yields a small denominator

and a large numerator, resulting in a high Gradient. Conversely, when the association is weak, the Gradient value is low. This makes the name "Domain Association Gradient" literal: it is the rate of change of performance with respect to FG-BG association strength [or association levels]. Our experimental procedure is described in detail in Section 3.

### 1.2 Contributions

- We highlight a crucial gap by quantifying the effect of context bias, suggesting that its mitigation is essential for enhancing model generalization and robustness across various environments. In contrast to existing approaches, we examine how context bias manifests across multiple domains in DAOD.
- We analyze FG–BG association and its underlying causal relationship using the drop rate and do-calculus. In addition, we employ distance-based metrics to measure FG-BG association under domain shifts.
- We propose a novel metric, *Gradient*, to quantify context bias in both source and target domains. To our knowledge, this is the first work to quantify this phenomenon.
- We provide a novel and practical research perspective by framing context bias as a critical factor in cross-domain object detection. Our study follows a logical progression from empirical observation to theoretical analysis, followed by quantitative and qualitative evaluations, resulting in strong evidence that FG–BG association significantly affects domain adaptation performance.

## 2 Related Work

### 2.1 FG–BG Association and Context Bias

There have been a number of studies aimed at improving performance in tasks such as classification, object recognition, and object localization, by examining the influence of background features (Xiao et al., 2020; Liang et al., 2023; Zhang et al., 2007; Ribeiro et al., 2016; Zhu et al., 2016; Rosenfeld et al., 2018; Barbu et al., 2019; Sagawa et al., 2019) and context bias (Torralba & Efros, 2011; Khosla et al., 2012; Choi et al., 2012; Shetty et al., 2019).

For example, Xiao et al. (2020) and Zhu et al. (2016) studied the effect of background on classification accuracy by modifying images with different combinations of foreground and background. Choi et al. (2012) proposed a graphical model that constructs FG–BG association using conditional probability and serves as a methodological inspiration for our work. Several studies have addressed context bias using techniques such as data augmentation to generate out-of-distribution samples in the background, combining naturally unmatched backgrounds and foregrounds (e.g., an elephant in a room), and applying background removal during training. Torralba (2003) demonstrated that background effects can be factorized into object priming, focus of attention, and scale selection by modeling FG–BG association within a probabilistic framework.

Liang et al. (2023) studied background influence using fashion datasets (Jia et al., 2020; Takagi et al., 2017). Other studies (Zhai et al., 2024; Wu et al., 2022) localized foreground objects more effectively than CAM-based algorithms, without bounding box information and using only classification labels.

These prior works focused on context bias within a single domain and used datasets with smaller variation, typically containing centered or isolated objects. Thus, a gap remains in understanding how context bias affects DAOD, particularly under domain shifts.

### 2.2 Domain Adaptation for Object Detection

Various DAOD methods have been proposed based on feature alignment, synthetic images, and self-training (or self-distillation). Feature alignment seeks transformations between source and target domains to reduce distribution shifts using adversarial training (He & Zhang, 2019; Chen et al., 2021; Ganin et al., 2016; Zhu et al., 2019), enabling the extraction of common latent features across domains. Progressive Domain Adaptation for Object Detection (Hsu et al., 2020) synthesized a new dataset using CycleGAN (Zhu et al.,

Table 1: Dataset and model abbreviations

| Abbreviation | Meaning |
|---|---|
| CST | Cityscapes Train |
| CSV/ CFV / CRV | Cityscapes Validation / Foggy / Rainy |
| KST | KITTI Semantic Train |
| BG-20K | Background 20K Dataset |
| VKC / VKF / VKM / VKO / VKR / VKS | Virtual KITTI Clone / Fog / Morning / Virtual KITTI Overcast / Rain / Sunset |
| ALDI++ | ResNet-50 FPN with ALDI++ best |
| Res | ResNet-50 FPN |
| Eff | EfficientNet-B0 FPN |
| Yo | YOLOv11 |

2017), to bridge domain gaps, while Self-Adversarial Disentangling for Specific Domain Adaptation (Zhou et al., 2023) achieved 45.2 mAP when adapting from the Cityscapes to Foggy Cityscapes dataset using synthetic images. Gong et al. (2022) utilized transformers to align features across backbone and decoder networks. However, combining multiple sources into a single dataset and performing single-source domain adaptation does not guarantee improved performance over using the best individual source domain (Zhao et al., 2020).

Self-training approaches use a teacher model to predict pseudo-labels in target domains, gradually accommodating domain shift (Caron et al., 2021; Pham et al., 2022; Cai et al., 2019; Chen et al., 2022; Cao et al., 2023). MIC (Hoyer et al., 2023) employed masked images in a teacher-student model, and MRT (Zhao et al., 2023) proposed a modified masking-based retraining approach for teacher-student models. Kay et al. (2024) combined alignment and distillation to enforce invariance across domains and reduce feature discrepancies.

Finding common features from multiple domains is critical for DAOD. These studies have demonstrated that foreground features in latent space can be aligned using dimension-reduction methods such as UMAP (McInnes et al., 2018) and t-SNE (Van der Maaten & Hinton, 2008). However, these studies do not explicitly address how to manage context bias during cross-domain adaptation. Instead, they have proposed and validated their methods within the DAOD framework using accuracy metrics. Therefore, we focus on analyzing the root causes of domain discrepancy in object detection, both qualitatively and quantitatively.

## 3 Method

Table 1 provides the abbreviations to refer to the datasets and models used throughout this paper.

### 3.1 Models

We employ ResNet-50 ("Res") and EfficientNet-B0 ("Eff") as backbones for FPN models implemented in Detectron2, as well as YOLOv11 ("Yo") (Khanam & Hussain, 2024), an anchor-free detection model. "Res" represents a backbone widely used across different architectures. "Eff" is chosen for its lightweight design. Additionally, we include the state-of-the-art DAOD method, ALDI++, which uses a ResNet-50 backbone, to evaluate its effectiveness in mitigating FG–BG association.

### 3.2 Datasets

We use multiple datasets for training and evaluation, including Cityscapes, KITTI Semantic, and various subsets of Virtual KITTI. Additionally, BG-20K, a collection of 20,000 images containing non-salient objects, is utilized to generate randomized background images. The Cityscapes and "KST" sets share 8 foreground and 11 background object categories, while the Virtual KITTI subsets contain 3 foreground and 10 background object classes.

The dataset sizes are as follows:

- **Cityscapes**: 2,950 training images, 500 validation images, 1,500 foggy validation images, and 1,188 rainy validation images.
- **KITTI Semantic Train**: 200 images.
- **Virtual KITTI Semantic**: 2,126 images across 6 simulated weather conditions. This dataset is synthetic and based on object tracking in diverse environments.

### 3.3 Training and Tests

We train "Res" and "Eff" on the "CST", "KST", and "VKC" datasets. "Yo" is trained with the same conditions using the Ultralytics framework. ALDI++ has been trained on ("KST", "CSV") and ("VKC", "VKF") as source and target domain pairs, respectively. We use the pre-trained ALDI++ model from the official repository without additional training on Cityscapes. For training, we use a learning rate of 0.02 for "Res", with input resolutions of 1024×2048 for Cityscapes and 375×1242 for KITTI-related datasets. For "Eff", we use a resolution of 1024×1024 for Cityscapes and the same KITTI resolution, with a learning rate of 0.01. All models use identical data augmentation: resizing and cropping, color jitter, and horizontal flipping. Each model is trained with a batch size of 8. We run training for approximately 100 epochs for ALDI++, "Res" and "Yo", and 200 epochs for "Eff". During evaluation, we use 1024×2048 images for Cityscapes-related datasets and 375×1242 for KITTI-related datasets on models except "Yo". "Yo" is trained and evaluated on 512x1024 and 320x1024 image resolutions respectively. The best model checkpoint is selected based on the highest mean Average Precision with 0.5 IoU threshold (mAP@50) from a DAOD perspective. For example, "Res" achieves a mAP@50 of 67.758 on the Cityscapes validation set and 54.617 on the "CFV" at epoch 7799, and 67.597 and 57.131 at epoch 7999, respectively. We select the model from epoch 7999 for subsequent experiments. For "KST", the model with the highest performance on "CFV" is chosen. For "VKC", we select the model with the highest mAP@50 on "VKF", which represents the largest domain shift among the Virtual KITTI variants. All models except "Yo" are trained using the standard loss functions provided by Detectron2, while Ultralytics is utilized for "Yo". Training is conducted using an NVIDIA RTX A4500 GPU.

### 3.4 Q1 - Exp1. Class-wise Background Removal Experiments in Image Space

The initial experiment evaluates the effect of background variation by performing inference on foreground objects placed over random background images fixed across different domains. Foreground regions are preserved and composited with backgrounds randomly sampled from the BG-20K dataset (see Figure 12 in Appendix A.1). The same sequence of background images is applied consistently across domains, thereby reducing the learned association between foreground and background. To ensure statistical validity, the experiment is repeated six times using different random sequences of background selections. Algorithm 1 describes the experiment process. The results, summarized using the mAP@50 metric and standard deviation, are presented in Table 4.

### 3.5 Q1-Exp2. Feature-wise Background Removal Experiments in Feature Space

The second experiment investigates feature-wise FG–BG association by selectively suppressing specific background labels in the feature space during inference. Using ground-truth semantic annotations, a particular background class (e.g., "road") is removed in repeated inference runs. This is achieved by zeroing out activation values in the corresponding background regions at shallow network layers: $res2.2$ in "Res" and ALDI++, $backbone.bottom_up._blocks.0$ in "Eff", and $model.1$ in "Yo". Due to the hierarchical nature of deep learning models, targeted suppression weakens FG–BG association for the removed background label, potentially affecting detection performance. To evaluate this effect, we measure the number of detections from unmodified models and compare them to detections after modification. The drop rate, calculated for each FG–BG class pair, reflects the sensitivity of foreground object detection to the presence of specific background labels.

**Definition of detection drop:** A detection is counted as dropped under any of the following conditions:

**Algorithm 1:** Class-wise Background Removal Experiments in Image Space

**Input:**

- $FG$: Set of foreground object instances
- $BG - 20K$: Set of 20,000 random background images
- $D$: Set of target domains for inference

**Output:** Mean and standard deviation of mAP@50 across 6 repeated trials

**foreach** *domain* $d \in D$ **do**
  **foreach** *foreground object* $f \in FG$ **do**
    Randomly sample a fixed set of background images $BG_i \subset BG_{20K}$;
    Synthesize image $I_{f,d}$ by placing $f$ onto a background from $BG_i$;
  Apply the trained model to all synthesized images $\{I_{f,d}\}$ for domain $d$;
  Measure detection performance using mAP@50;
Compute mean and standard deviation of mAP@50 across all six trials;

**Algorithm 2:** Feature-wise Background Removal Experiments in Feature Space

**Input:**

- $FG$: Set of foreground object instances
- $BG$: Set of background regions with semantic labels
- $L_{remove}$: Background label to be removed (e.g., "road")
- $M$: Deep learning models (e.g., "Res")

**Output:** Drop rate statistics for each FG–BG pair and Wilcoxon test results

**for** $i = 1$ **to** 6 **do**
  **foreach** *model* $m \in M$ **do**
    **foreach** *image x with semantic ground truth* **do**
      Perform standard inference on $x$ with model $m$, store number of detections $D_{std}$;
      Remove BG activated pixels in shallow feature maps corresponding to label $L_{remove}$ (e.g., `res2.2` for "Res");
      Perform modified inference on $x$, store number of detections $D_{mod}$;
      Compute detection drop $\Delta D = D_{std} - D_{mod}$ for each FG–BG pair;
  Store all $\Delta D$ values for statistical analysis;
Aggregate drop rates for each FG–BG pair across all trials;
Conduct a Wilcoxon signed-rank test to assess the significance of the drop rate distributions;

1. The predicted class changes due to background label removal.
2. The defined loss of information exceeds 1.0, meaning the IoU with ground truth significantly decreases (less than 0.1 IoU).
3. The prediction matches a different ground-truth object not originally considered a true positive.

Importantly, drop rates are computed only for true positive cases. To ensure statistical rigor, the process is repeated six times. Since drop rate distributions do not satisfy normality assumptions, we employ the Wilcoxon signed-rank test (Woolson, 2005) to assess statistical significance. Algorithm 2 describes the experiment process.

### 3.5.1 Q2-Exp1. FG–BG Association with Respect to Activated Background Region

Using CAM masks with varying thresholds, we measure the mAP@50 drop rate to investigate the causal influence of activated background regions on object detection performance. Smooth-GradCAM++ (Omeiza et al., 2019) generates contextually meaningful instance masks by computing gradients with respect to each object's score, using a 0.85 confidence threshold. The CAM masks are binarized by applying progressively lower threshold values, decreasing by 0.1 with each bin. The extent of the activated background region is controlled by the chosen threshold, while the masked foreground region remains fixed throughout the experiment, regardless of background variation. Statistical analysis demonstrates the causality between association and accuracy. Algorithm 3 describes the experimental process. Figure 7 illustrates the contextual masks across different layers and bins.

We define the hit ratio as the number of foreground and background pixels captured by CAM in the activation maps, normalized by the number of ground truth pixels in the instance masks. "FG mean" is calculated as the average number of pixels captured both by CAM and the instance mask ground truth. "BG mean" is the average ratio of activated background pixels to foreground pixels. We average all instances' hit ratios to determine the "FG mean" and "BG mean". This indicates that CAM appropriately captures contextual information for each instance. The definitions of associated and non-associated features are provided in Section 3.6.1.

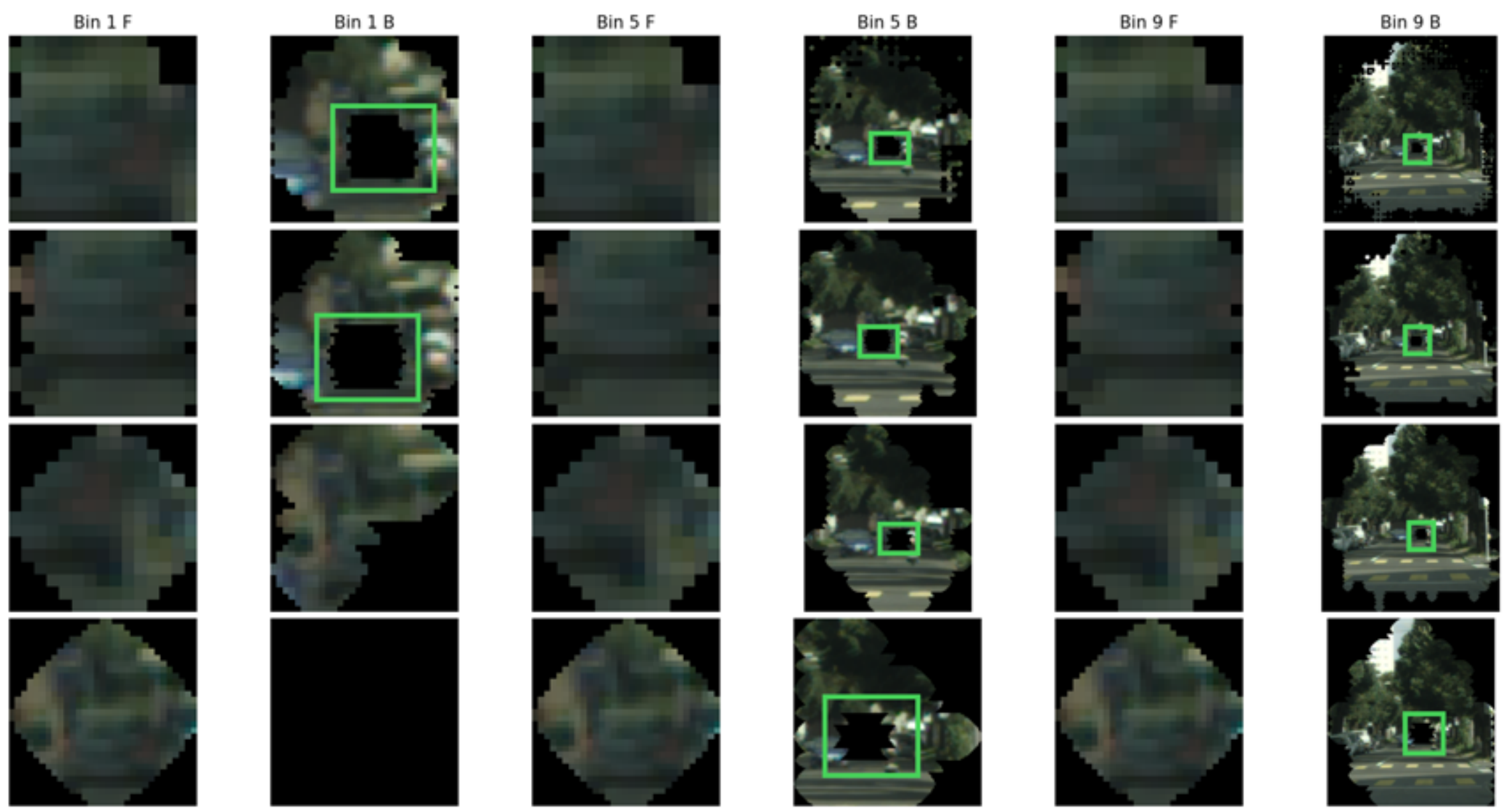

Figure 7: **"Res" foreground and background images using CAM and instance masks.** Each column corresponds to bins 1, 5, and 9 with foreground and background orders. Each row is a different layer. The green bounding boxes highlight the location of the foreground object. Foreground regions are maintained across layers. Image blurriness is due to scaling.

## 3.6 Q3-Exp1 and Exp2. Quantifying the Causal Effect on DAOD

The results from our experiments confirm the existence and causal effect of FG–BG association. However, these findings do not directly quantify their influence on DAOD. To address this, we introduce a new metric, Domain Association Gradient (referred to as *Gradient*), which measures performance perturbation in response to the strength of FG–BG association. To quantify association strength, we add the MMD between FG and BG (`f2b` and `b2f`) and subtract the MMD measured within FG and within BG (`f2f` and `b2b`). Figure 8 illustrates the feature extraction breakdown. For each domain ($D$), we separate the features into associated ($F_a^D$) and non-associated ($F_{na}^D$) features. These features can then be further broken down into

---

**Algorithm 3:** Causality Analysis via Smooth-GradCAM++ Mask Thresholding

**Input:**

- $FG$: Set of foreground object instances with 0.85 confidence threshold
- $x$: Input image
- $T$: Set of CAM thresholds (e.g., maximum of activation value to $1e^{-9}$ in 0.1 decrements)

**Output:** Drop rates under different CAM thresholds

**foreach** *foreground instance* $f \in FG$ **do**
    Generate Smooth-GradCAM++ map $H_f$ using prediction confidence of $f$ from image $x$;
    **foreach** *threshold* $t \in T$ **do**
        Binarized CAM mask: $M_f^{(t)} = \mathbb{1}(H_f \geq t)$;
        *(Note: foreground region remains fixed; only background area changes)*;
        Remove partial BG activations in shallow feature maps corresponding to the Masks (e.g., $res2.2$ for "Res") and compute detection result $D_{mod}$;
        Compute drop rate: $\Delta D_t = D_{std} - D_{mod}$ where $D_{std}$ is detection without CAM masking;
Aggregate drop rates and hit ratios across all instances;
Analyze drop rate trend across thresholds to infer FG–BG causality;

---

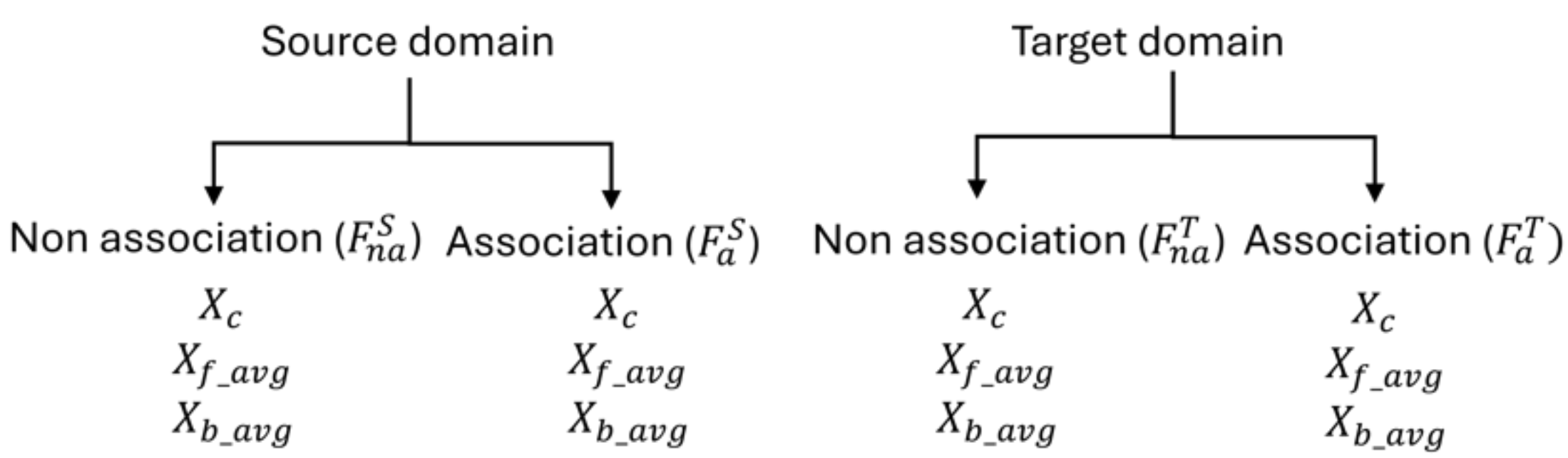

Figure 8: **Feature extraction graph.** We extract features from all true positives with different FG–BG pairs. The process for generating each feature is described in Algorithm 4.

CAM-activated features ($X_c$), 2D pooled average FG features within $X_c$ ($X_{f_{avg}}$), and 2D pooled average BG features within $X_c$ ($X_{b_{avg}}$), using instance masks to distinguish FG and BG features. The detailed extraction process for features such as $X_c$, $X_{f_{avg}}$, and $X_{b_{avg}}$ is described in Section 3.6.1.

The intuition is that if a strong FG–BG association is learned, the MMD between features preserving this association should be small due to shared contextual dependencies, while the drop rate should be large. Consequently, *Gradient* values computed from these features should exceed those computed from features lacking FG–BG association. We calculate *Gradient* values for each FG–BG pair using the drop rate and analyze their patterns using Equation 5. Only FG–BG pairs with statistically significant drop rate differences identified in Experiment 3.5 are included in the analysis. Based on our hypothesis, we expect $\text{Gradient}^S$ to exceed $\text{Gradient}^T$, as models trained on the source domain tend to capture stronger FG–BG associations. Superscripts $S$ and $T$ denote source and target domains, respectively.

$$\text{Gradient}^{\text{S}} = \frac{\text{Source domain drop rate}}{\text{f2b}^{\text{S}} + \text{b2f}^{\text{S}} - \text{f2f}^{\text{S}} - \text{b2b}^{\text{S}}}, \quad \text{Gradient}^{\text{T}} = \frac{\text{Target domain drop rate}}{\text{f2b}^{\text{T}} + \text{b2f}^{\text{T}} - \text{f2f}^{\text{T}} - \text{b2b}^{\text{T}}} \tag{5}$$

Equation 6 defines how various feature combinations, drawn from different contexts, are used in computing this metric. Specifically, cross-context feature comparisons, such as `f2b` and `b2f`, are expected to yield higher

MMD values than within-context comparisons like `f2f` and `b2b`, since the latter reflect similar contextual structures.

$$
\begin{aligned}
\mathrm{f2f}^{\mathrm{S}} &= \mathrm{MMD}(X_{f_avg} \in F^S_{na},\, X_{f_avg} \in F^S_{a}) & \mathrm{f2f}^{\mathrm{T}} &= \mathrm{MMD}(X_{f_avg} \in F^T_{na},\, X_{f_avg} \in F^T_{a}) \\
\mathrm{f2b}^{\mathrm{S}} &= \mathrm{MMD}(X_{f_avg} \in F^S_{na},\, X_{b_avg} \in F^S_{a}) & \mathrm{f2b}^{\mathrm{T}} &= \mathrm{MMD}(X_{f_avg} \in F^T_{na},\, X_{b_avg} \in F^T_{a}) \\
\mathrm{b2f}^{\mathrm{S}} &= \mathrm{MMD}(X_{b_avg} \in F^S_{na},\, X_{f_avg} \in F^S_{a}) & \mathrm{b2f}^{\mathrm{T}} &= \mathrm{MMD}(X_{b_avg} \in F^T_{na},\, X_{f_avg} \in F^T_{a}) \\
\mathrm{b2b}^{\mathrm{S}} &= \mathrm{MMD}(X_{b_avg} \in F^S_{na},\, X_{b_avg} \in F^S_{a}) & \mathrm{b2b}^{\mathrm{T}} &= \mathrm{MMD}(X_{b_avg} \in F^T_{na},\, X_{b_avg} \in F^T_{a})
\end{aligned} \tag{6}
$$

Furthermore, we conduct a focused analysis on `f2b` and `b2f` MMD values for both associated and non-associated features across domains using equation 7 and 8. We compute the sum of these values and compare them using a pairwise T-test (O'Mahony, 2017) after the Shapiro-Wilk test (Shapiro & Wilk, 1965).

$$
\begin{aligned}
f2b_{asso} &= \mathrm{MMD}(X_{f_avg} \in F^S_{a},\, X_{b_avg} \in F^T_{a}) & f2b_{no-asso} &= \mathrm{MMD}(X_{f_avg} \in F^S_{na},\, X_{b_avg} \in F^T_{na}) \\
b2f_{asso} &= \mathrm{MMD}(X_{b_avg} \in F^S_{a},\, X_{f_avg} \in F^T_{a}) & b2f_{no-asso} &= \mathrm{MMD}(X_{b_avg} \in F^S_{na},\, X_{f_avg} \in F^T_{na})
\end{aligned} \tag{7}
$$

$$
\mathrm{sum}_{\text{a-a}} = \mathrm{f2b}_{\text{asso}} + \mathrm{b2f}_{\text{asso}} \qquad \mathrm{sum}_{\text{na-na}} = \mathrm{f2b}_{\text{no-asso}} + \mathrm{b2f}_{\text{no-asso}} \tag{8}
$$

A higher $sum_{na-na}$ than $sum_{a-a}$ indicates that FG–BG association persists across domains and may influence DAOD performance when the model unintentionally leverages such dependencies during inference. In other words, if FG–BG association does not impact DAOD, then $sum_{na-na}$ and $sum_{a-a}$ should not show a statistical difference.

We randomly sample non-associated features to match the number of associated features. We divide all datasets into two groups: the first group includes Cityscapes-related datasets and "KST", while the second group contains only the Virtual KITTI-related dataset, due to unmatched label configurations.

#### 3.6.1 Associated and Non-associated Feature Extraction

With the drop rate experiment (Algorithm 3) as pre-processing, we define associated features as features extracted from instances that are not detected without background context, indicating that the features encode the association between FG and BG. Conversely, non-associated features indicate that the FG-BG association is absent from the extracted features. We extract and cluster features per FG–BG pair. We use four layers per model to extract features at different scales. For "Res" and ALDI++, we use $res2.2, res3.3, res4.5, res5.2$ and for "Eff", we use $backbone.block.0, 1, 3$, and 5. From each domain, we store both associated and non-associated features. Algorithm 4 describes steps to process foreground-related features and background-related features. Table 2 defines $F^D_a$ and $F^D_{na}$, as FG–BG associated and non-associated features from each domain $D$.

## 4 Experiments

### Q1. Is FG–BG association being inadvertently learned during the training process?

### Model Evaluation

We evaluated the trained models using the mAP@50 metric. Table 3 summarizes the evaluation results. Among models trained on the Cityscapes dataset, ALDI++ outperformed others on Cityscapes-related datasets and "KST", likely due to longer training on the target domain and the use of multiple domain datasets via DAOD algorithms. Although "Yo" achieved strong performance on "CST", its domain adaptation capability was weaker than that of the baseline "Res". "Eff" consistently showed the lowest performance on both Cityscapes and "KST", but outperformed "Yo" on "KST". For models trained on "KST", ALDI++ again outperformed other models, while "Eff" and "Yo" showed significantly worse results. The relatively small "KST" dataset, with only 200 images, introduced a domain shift that limited generalization when models were transferred to larger datasets. Consequently, "Res" and "Yo" also exhibited weakened performance due to insufficient data. With VKC-trained models, "Res" and "Yo" demonstrated reasonable domain

**Algorithm 4:** Feature Extraction from CAM and Ground Truth Instance Mask

**Input:**
- $C$: CAM mask from Algorithm 3.
- $G$: Ground truth instance binary mask
- $A$: Activation maps from different layers (e.g. $res2.2, res3.3, res4.5$, and $res5.2$ for "Res")

**Output:** $X_c$, $X_{f_avg}$, $X_{b_avg}$

Compute features $X$ from CAM mask $C$;
    $X = A \cdot \mathbb{1}(C = 1)$;
Compute Normalized features $X_c$ from $X$;
    $X_c =$Normalize($X$);
    Separate normalized features using ground truth mask $G$:
    $X_{f_avg} =$ Adaptive pool 2d($X_c \cdot \mathbb{1}(G = 1)$);
    $X_{b_avg} =$ Adaptive pool 2d($X_c \cdot \mathbb{1}(G = 0)$);

Table 2: **Definition of FG–BG associated and non-associated features at each domain $D$.** "1" indicates that BG removal is performed and detection fails and "0" indicates BG removal is not applied and detection succeeds.

| | | **Detection** | |
|---|---|---|---|
| | | **0** | **1** |
| **BG removal** | **1** | When "road" is removed, detection fails → car feature with association ($F_a^D$) | Without "road", detection succeeds → car feature without association ($F_{na}^D$) |
| | **0** | False Negative. Unknown association impact on prediction. | True Positive. No association impact on prediction. |

generalization compared to "Eff". However, on the "VKF" validation set, "Yo"'s performance dropped significantly relative to "Res", despite its overall strong results on other datasets. "Eff" also experienced a notable performance decline on "VKF". Overall, ALDI++ demonstrated the effectiveness of DAOD-based domain generalization methods.

#### Q1-Exp1. Class-wise Background Removal Experiments in Image Space

This experiment evaluated the role of FG–BG association by replacing background regions with images containing non-salient objects while preserving the foreground objects. We measured mAP@50 over six repetitions using randomly generated images. Table 4 presents the mean and standard deviation across these six evaluation runs. Models trained on Cityscapes showed substantial performance drops compared to their baseline evaluation. Notably, ALDI++ exhibited greater degradation than "Res" on "KST", suggesting that ALDI++ relies heavily on FG–BG association learned from both the source and target domains. Similarly, models trained on "KST" and "VKC" also experienced considerable performance declines, indicating that FG–BG association was consistently learned during training and exploited by the models during inference.

#### Q1-Exp2. Feature-wise Background Removal Experiments in Feature Space

In addition to background perturbation in image space, we performed background removal in feature space as shown in Table 5. Specifically, we zeroed out activated background regions in the shallow layers of each model architecture, preventing background information from propagating to deeper layers. This effectively disables the FG–BG association during inference. The Cityscapes-related datasets and "KST" contain 88 distinct FG–BG combinations, while the Virtual KITTI datasets include 30 combinations. We report only the combinations that resulted in a statistically significant performance drop of at least 8%. Additional

Table 3: Model evaluation across different training and validation sets. "-" indicates cases that are not measurable due to unmatched class categories.

**Cityscapes Trained**

| Dataset | Res | Eff | Yo | ALDI++ |
|---|---|---|---|---|
| CST | 79.14 | 41.12 | **88.26** | 87.97 |
| CSV | 67.59 | 42.90 | 59.56 | **70.08** |
| CFV | 57.13 | 20.58 | 44.49 | **67.45** |
| CRV | 58.65 | 23.18 | 48.77 | **69.78** |
| KST | 46.25 | 28.92 | 23.91 | **47.96** |

**KST Trained**

| Dataset | Res | Eff | Yo | ALDI++ |
|---|---|---|---|---|
| CST | - | - | - | - |
| CSV | 43.23 | 2.74 | 21.69 | **51.09** |
| CFV | 35.48 | 0.61 | 12.53 | **43.62** |
| CRV | 37.42 | 0.86 | 17.57 | **47.82** |
| KST | 86.17 | 10.42 | 21.53 | **92.44** |

**VKC Trained**

| Dataset | Res | Eff | Yo | ALDI++ |
|---|---|---|---|---|
| VKC | 81.67 | 50.09 | **85.99** | 81.96 |
| VKF | 61.14 | 5.80 | 34.27 | **72.60** |
| VKM | 79.72 | 29.52 | 79.55 | **80.27** |
| VKO | 75.14 | 30.41 | **81.93** | 78.58 |
| VKR | 71.66 | 25.53 | 75.93 | **78.36** |
| VKS | 76.02 | 26.18 | **78.65** | 77.59 |

combinations exhibited smaller drops and are therefore not included. Figure 9 illustrates an example of a significant performance drop on "CST" using the "Res" model. These result indicate that the models learned notable FG–BG associations during training, which can cause performance degradation.

### Q2. Is there a causal relationship between FG–BG association and object detection?

In the previous section, we validated the existence of FG–BG association. To further investigate this phenomenon, we computed CAM masks for each object and analyzed background-region-based association. To explore causality, we applied the do-calculus using CAM-derived masks and ground-truth instance masks. We also note that "Yo" contains non-differentiable non-maximum suppression (NMS), which prevents the use of gradient-based CAM to capture contextual masks for specific objects. Thus, we are unable to conduct the Q2 and Q3 experiments using this model.

### Q2-Exp1. FG–BG Association with Respect to Activated Background Regions

With do-calculus, we computed the mean drop rate across all classes for each bin. While foreground regions were maintained regardless of bin (see Fig. 7 and Table 6), the drop rate significantly increased with bin 1, which contains a small amount of activated background (see Table 7). As the activated background region expanded, the drop rate converged toward zero, indicating correct object detection. Through this experiment, we confirmed a causal relationship between FG–BG association and detection outcomes, as the accuracy of foreground objects changed notably across bins, particularly between bin 1 to bin 5.

### Q3. What is the impact of FG–BG association on DAOD and how can the effect be quantified?

We computed *Gradient* values and analyzed the MMDs between associated and non-associated features across different domains. This allowed us to quantify the impact of FG–BG association on domain shifts.

Table 4: **Mean ± standard deviation of mAP@50 across synthetic datasets and models trained on Cityscapes, "KST", and "VKC".** Bolded values indicate the highest mAP@50 for each training-dataset-model pair.

**Cityscapes Trained**

| **Dataset + BG** | Res | Eff | Yo | ALDI++ |
|---|---|---|---|---|
| CSV | 44.5 ± 6.1 | 10.02 ± 6.4 | 32.78 ± 0.6 | **47.96** ± 6.3 |
| CFV | 34.6 ± 10.6 | 8.99 ± 4.5 | 15.70 ± 0.3 | **43.17** ± 9.8 |
| CRV | 29.1 ± 4.4 | 14.24 ± 8.4 | 17.37 ± 0.3 | **38.18** ± 5.1 |
| KST | **41.4** ± 4.4 | 25.33 ± 6.4 | 19.92 ± 1.4 | 33.98 ± 6.7 |

**KST Trained**

| **Dataset + BG** | Res | Eff | Yo | ALDI++ |
|---|---|---|---|---|
| CSV | 19.2 ± 5.2 | 0.16 ± 0.1 | 21.34 ± 0.3 | **23.01** ± 0.7 |
| CFV | 17.0 ± 5.8 | 0.22 ± 0.1 | 10.35 ± 0.1 | **24.55** ± 0.1 |
| CRV | 13.6 ± 3.4 | 2.25 ± 2.1 | 14.29 ± 0.1 | **35.29** ± 0.7 |

**VKC Trained**

| **Dataset + BG** | Res | Eff | Yo | ALDI++ |
|---|---|---|---|---|
| VKF | 38.7 ± 10.7 | 1.17 ± 0.5 | 14.61 ± 0.5 | **51.04** ± 0.5 |
| VKM | 61.8 ± 5.1 | 11.44 ± 6.4 | 35.26 ± 0.6 | **64.17** ± 0.3 |
| VKO | 60.0 ± 4.7 | 10.70 ± 5.5 | 33.66 ± 0.4 | **64.44** ± 0.5 |
| VKR | 58.4 ± 3.5 | 7.99 ± 3.5 | 27.56 ± 0.4 | **59.81** ± 0.4 |
| VKS | 59.8 ± 6.5 | 12.92 ± 4.6 | 35.93 ± 0.8 | **64.26** ± 0.2 |

Table 5: **Statistically significant differences in the number of FG-BG pairs across models**. Only FG–BG pairs with a drop rate greater than 8% are reported. Bolded values indicate stronger FG–BG association for each model on the corresponding datasets.

| | **Res** | **ALDI++** | **Eff** | **Yo** |
|---|---|---|---|---|
| CST | 14/88 | **18/88** | 15/88 | 12/88 |
| CSV | 7/88 | **17/88** | 7/88 | 3/88 |
| CFV | 13/88 | **20/88** | 11/88 | 7/88 |
| CRV | 15/88 | **21/88** | 12/88 | 6/88 |
| KST | 2/88 | 2/88 | **4/88** | 2/88 |
| VKC | 8/30 | 7/30 | **9/30** | 4/30 |
| VKF | **7/30** | **7/30** | 4/30 | 2/30 |
| VKM | **9/30** | 7/30 | 7/30 | 3/30 |
| VKO | **9/30** | 7/30 | 8/30 | 3/30 |
| VKR | 7/30 | **9/30** | **9/30** | 6/30 |
| VKS | 8/30 | 8/30 | **9/30** | 4/30 |

### Q3-Exp1. Domain Association Gradient

To validate our hypothesis that $Gradient^S$ should be higher than $Gradient^T$ due to learned FG–BG association, we categorized the comparisons into three cases: **(1) $\boldsymbol{Gradient^S}$ significantly higher than $\boldsymbol{Gradient^T}$, (2) $\boldsymbol{Gradient^S}$ significantly lower than $\boldsymbol{Gradient^T}$**, and **(3) no statistically significant difference.** Table 8 summarizes the results. Overall, the findings support our hypothesis. However, ALDI++ showed opposite results on the CST-CSV pair, possibly due to the DAOD training strategy using target domain information (CFV). In the Virtual KITTI-related datasets, particularly for "Eff", some results contradicted expectations. This may be attributed to strong spatial and temporal correlations inherent in the dataset, which is derived from object-tracking video sequences, or to biases introduced by a small number

Table 6: **Definition of FG–BG associated and non-associated features for each domain D.** The FG mean of 1.0 indicates that all foreground pixels are captured by the CAM. The BG mean represents the ratio of captured background pixels to foreground pixels, reflecting the extent of background activation relative to the foreground.

| Layer | Hit ratio | | | |
|---|---|---|---|---|
| | Associated | | Non-associated | |
| | FG mean | BG mean | FG mean | BG mean |
| $res2.2$ | 1.0 | 14.81 | 1.0 | 20.04 |
| $res3.3$ | 1.0 | 14.33 | 1.0 | 18.79 |
| $res4.5$ | 1.0 | 12.66 | 1.0 | 27.2 |
| $res5.2$ | 1.0 | 21.0 | 1.0 | 12.33 |

Table 7: **Drop rate by bin.** For each model (Res, ALDI++, Eff), columns B1, B5, and B9 represent increasing levels of activated background. Each entry reports the drop rate for a given dataset and bin. Lower values indicate fewer or no detection drops. Bolded values highlight notable performance degradation.

| | **Res** | | | **ALDI++** | | | **Eff** | | |
|---|---|---|---|---|---|---|---|---|---|
| | B1 | B5 | B9 | B1 | B5 | B9 | B1 | B5 | B9 |
| CST | **0.65** | 0.02 | 0.00 | **0.66** | 0.02 | 0.00 | **0.74** | 0.25 | 0.18 |
| CSV | **0.62** | 0.02 | 0.00 | **0.64** | 0.01 | 0.00 | **0.71** | 0.14 | 0.10 |
| CFV | **0.67** | 0.03 | 0.00 | **0.68** | 0.02 | 0.00 | **0.79** | 0.09 | 0.11 |
| CRV | **0.67** | 0.07 | 0.01 | **0.64** | 0.05 | 0.01 | **0.65** | 0.06 | 0.09 |
| KST | **0.84** | 0.22 | 0.00 | **0.70** | 0.00 | 0.02 | **0.82** | 0.36 | 0.68 |
| VKC | **0.57** | 0.02 | 0.00 | **0.33** | 0.01 | 0.00 | **0.71** | 0.07 | 0.03 |
| VKF | **0.75** | 0.12 | 0.01 | **0.42** | 0.02 | 0.00 | **0.92** | 0.04 | 0.02 |
| VKM | **0.63** | 0.03 | 0.00 | **0.42** | 0.02 | 0.00 | **0.77** | 0.11 | 0.05 |
| VKO | **0.56** | 0.03 | 0.00 | **0.33** | 0.02 | 0.00 | **0.76** | 0.05 | 0.05 |
| VKR | **0.62** | 0.04 | 0.00 | **0.33** | 0.02 | 0.00 | **0.80** | 0.11 | 0.07 |
| VKS | **0.64** | 0.07 | 0.01 | **0.34** | 0.02 | 0.00 | **0.77** | 0.06 | 0.03 |

of detections and drops. These results elaborate on the quantification of class-wise causal effects in DAOD. Figure 10 presents the results of the *Gradient* comparisons as box plots for visual comparison.

#### Q3-Exp2. Associated and Non-associated Features

To understand the class- and feature-wise impact across different domains, we compared the gap between the MMD of associated features and that of non-associated features. Similar to the *Gradient* analysis, we categorized comparisons into three cases: **(1) The summation of MMDs (`f2b` and `b2f`) of associated features significantly lower than that of non-associated features (2) significantly higher**, and **(3) no statistically significant difference.** Overall, associated features sharing the same FG–BG association across domains exhibited lower MMD than non-associated features, indicating stronger cross-domain FG-BG association consistency in cross-domain. However, "Eff" showed reversed outcomes on the CST-CRV and CST-KST pairs. This may be due to a limited number of detections, resulting in insufficient feature representations or overall poor model performance. Table 9 presents the results, while Figure 11 depicts the results of associated and non-associated feature comparisons with statistical annotations.

## 5 Discussion and Conclusion

In this work, we present a comprehensive empirical and theoretical investigation into the role of context bias in DAOD. While context bias has previously been studied in classification and segmentation tasks, our work is the first to formally identify, quantify, and causally analyze this phenomenon within the context of DAOD.

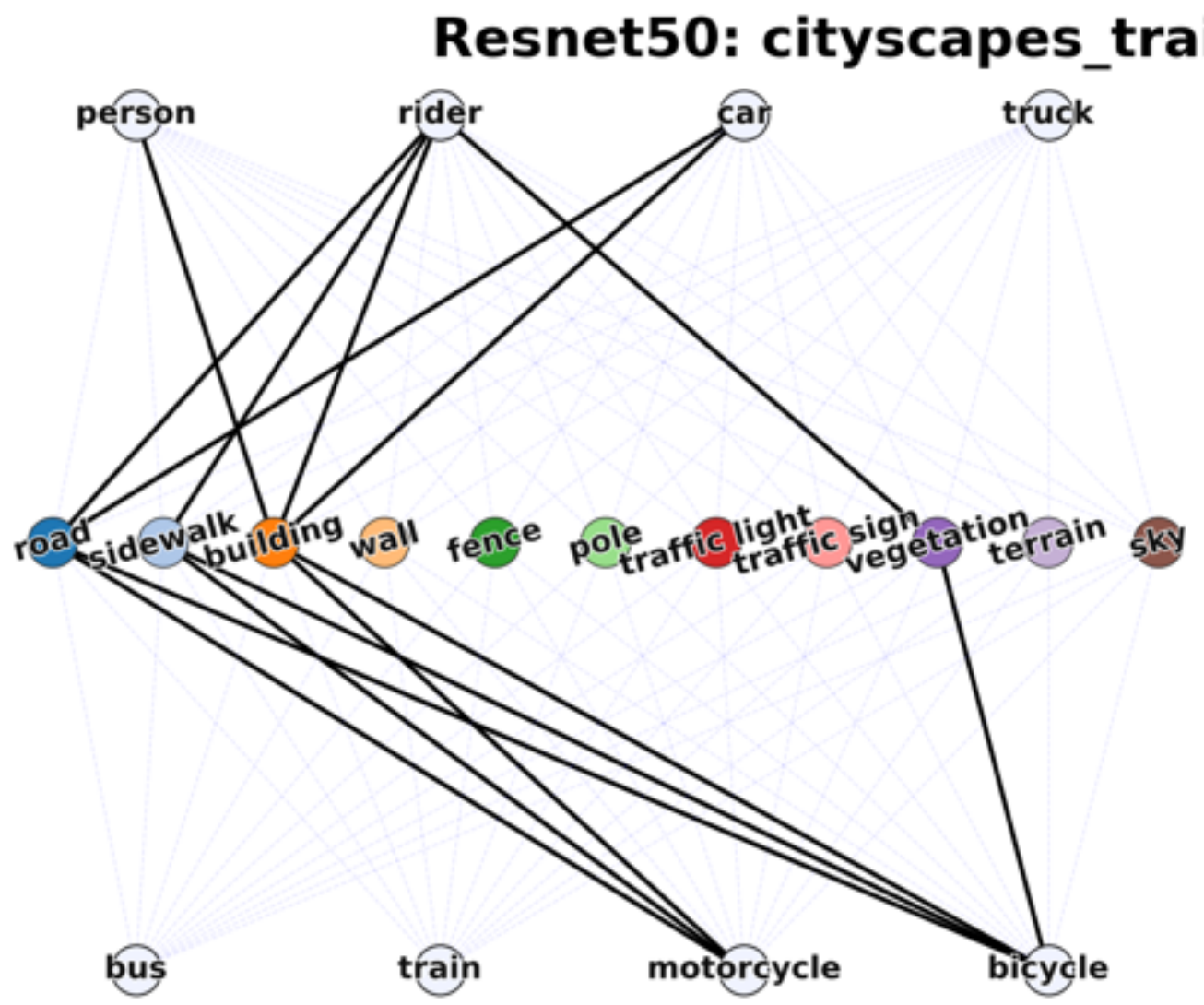

| BG | FG | Drop rate |
|---|---|---|
| building | person | 0.22 |
| road | rider | 0.21 |
| sidewalk | bicycle | 0.21 |
| building | bicycle | 0.20 |
| building | rider | 0.19 |
| building | motorcycle | 0.18 |
| road | bicycle | 0.18 |
| road | motorcycle | 0.16 |
| sidewalk | motorcycle | 0.16 |
| sidewalk | rider | 0.14 |
| vegetation | rider | 0.12 |
| building | car | 0.11 |
| road | car | 0.10 |
| vegetation | bicycle | 0.09 |

Figure 9: **Feature-wise drop rates of "Res" on "CST".** Bold edges indicate FG–BG pairs with statistically significant differences. The table on the right in the figure illustrates drop rates in decreasing order.

Table 8: *Gradient* **comparison across different domains.** The denominator of each entry indicates the number of FG–BG associations shared across the two domains. Cases 1, 2, and 3 are denoted as C1, C2, and C3, respectively. Bold values represent the dominant case for each dataset-model pair.

| | **Res** | | | **ALDI++** | | | **Eff** | | |
|---|---|---|---|---|---|---|---|---|---|
| | C1 | C2 | C3 | C1 | C2 | C3 | C1 | C2 | C3 |
| CST - CSV | **4/5** | 0/5 | 1/5 | 4/10 | **6/10** | 0/10 | **3/7** | **3/7** | 1/7 |
| CST - CFV | **11/11** | 0/11 | 0/11 | **15/15** | 0/15 | 0/15 | **7/11** | 2/11 | 2/11 |
| CST - CRV | **10/11** | 1/11 | 0/11 | **8/10** | 2/10 | 0/10 | **7/10** | 3/10 | 0/10 |
| CST - KST | **1/1** | 0/1 | 0/1 | **1/1** | 0/1 | 0/1 | **2/3** | 1/3 | 0/3 |
| VKC - VKF | **5/5** | 0/5 | 0/5 | **4/4** | 0/4 | 0/4 | **1/2** | **1/2** | 0/2 |
| VKC - VKM | **5/7** | 2/7 | 0/7 | **4/4** | 0/4 | 0/4 | 1/5 | **4/5** | 0/5 |
| VKC - VKO | **4/7** | 3/7 | 0/7 | **4/4** | 0/4 | 0/4 | 1/5 | **4/5** | 0/5 |
| VKC - VKR | **6/6** | 0/6 | 0/6 | **5/5** | 0/5 | 0/5 | 1/5 | **4/5** | 0/5 |
| VKC - VKS | 2/6 | **4/6** | 0/6 | **4/5** | 1/5 | 0/5 | 1/5 | **4/5** | 0/5 |

Our findings show that modern object detection models frequently rely on FG–BG association, which often does not generalize well across domains. Through systematic background masking and feature-level perturbations, we demonstrate that removing or altering background information can lead to substantial drops in detection performance even when the foreground remains intact. These effects are consistent across multiple model architectures and domain pairs, including ALDI++.

Furthermore, we show that FG–BG association is not only empirically observable but also causally linked to detection outcomes. Using a combination of do-calculus, Smooth-GradCAM++, and layer-wise feature analysis, we construct and validate a causal model that quantifies the influence of context bias. We introduce a novel metric and find that domain shifts exacerbate performance disparities when models rely on FG–BG association.

In the Appendix (Section A.3), we provide additional clarifications of our experimental results. Specifically, we include (i) a fixed color background replacement experiment analyzed from a counterfactual perspective; (ii) ViTDet (Li et al., 2022a) attention maps and drop rates under background replacement in image space; and (iii) *Gradient*, and a comparison of associated versus non-associated features using a Radial Basis

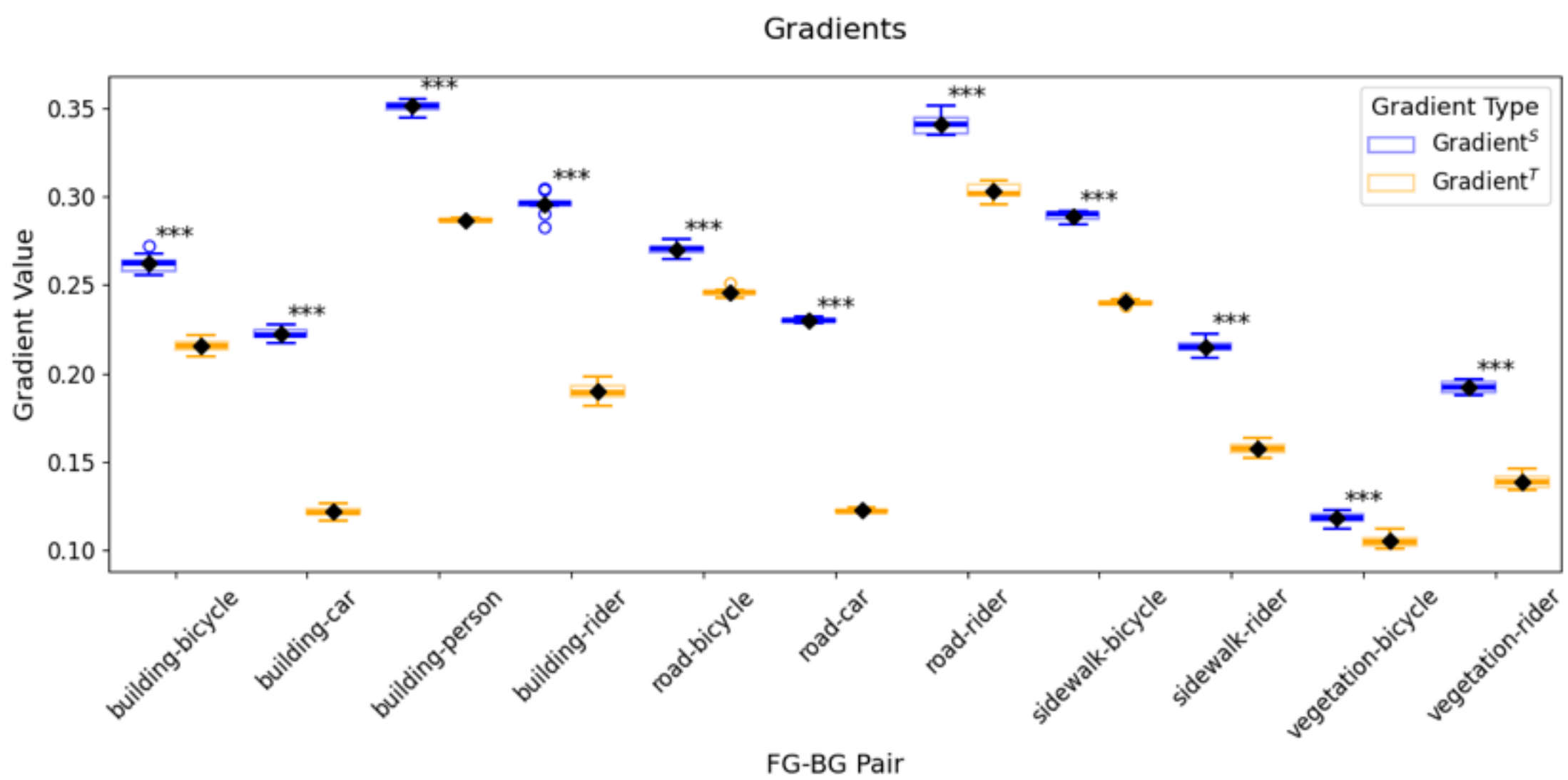

Figure 10: ***Gradient$^S$* and *Gradient$^T$*comparison of "Res" model with "CST" and "CFV"**. The number of asterisks ("*") indicates the level of statistical significance of a p-value.

Table 9: **Comparison of associated and non-associated features across different domains.** The denominator of each entry indicates the number of FG–BG associations shared across the two domains. Cases 1, 2, and 3 are given as C1, C2, and C3. Bold values represent the dominant case for each dataset-model pair.

| | **Res** | | | **ALDI++** | | | **Eff** | | |
|---|---|---|---|---|---|---|---|---|---|
| | C1 | C2 | C3 | C1 | C2 | C3 | C1 | C2 | C3 |
| CST - CSV | **5/5** | 0/5 | 0/5 | **9/10** | 0/10 | 1/10 | **3/7** | 2/7 | 2/7 |
| CST - CFV | **11/11** | 0/11 | 0/11 | **9/15** | 6/15 | 0/15 | **6/11** | 5/11 | 0/11 |
| CST - CRV | **8/11** | 3/11 | 0/11 | **7/10** | 3/10 | 0/10 | 2/10 | **6/10** | 2/10 |
| CST - KST | **1/1** | 0/1 | 0/1 | **1/1** | 0/1 | 0/1 | 0/3 | **2/3** | 1/3 |
| VKC - VKF | **2/5** | 1/5 | **2/5** | **4/4** | 0/4 | 0/4 | **2/2** | 0/2 | 0/2 |
| VKC - VKM | **4/7** | 2/7 | 1/7 | **4/4** | 0/4 | 0/4 | **3/5** | 1/5 | 1/5 |
| VKC - VKO | **5/7** | 2/7 | 0/7 | **4/4** | 0/4 | 0/4 | **4/5** | 1/5 | 1/5 |
| VKC - VKR | **4/6** | 1/6 | 1/6 | **5/5** | 0/5 | 0/5 | **4/5** | 1/5 | 0/5 |
| VKC - VKS | **5/6** | 1/6 | 0/6 | **5/5** | 0/5 | 0/5 | **3/5** | 1/5 | 1/5 |

Function (RBF) kernel. Together, these supplementary analyses help address several open questions and limitations of our main experiments.

## Limitations

Despite the strength of our analysis, we acknowledge that extracting foreground and background features separately across large datasets is computationally expensive. This limits the scalability of some of the proposed methods. Some outliers may result from class imbalance among foreground objects across the datasets. For example, "car" and "person" are dominant but other foreground objects are rare. There are also certain neural network architectures, such as in "Yo", that prevent us from computing the CAM masks and deriving the FG and BG activation features. Thus, our approach is limited to models that support gradient-based CAM methods.

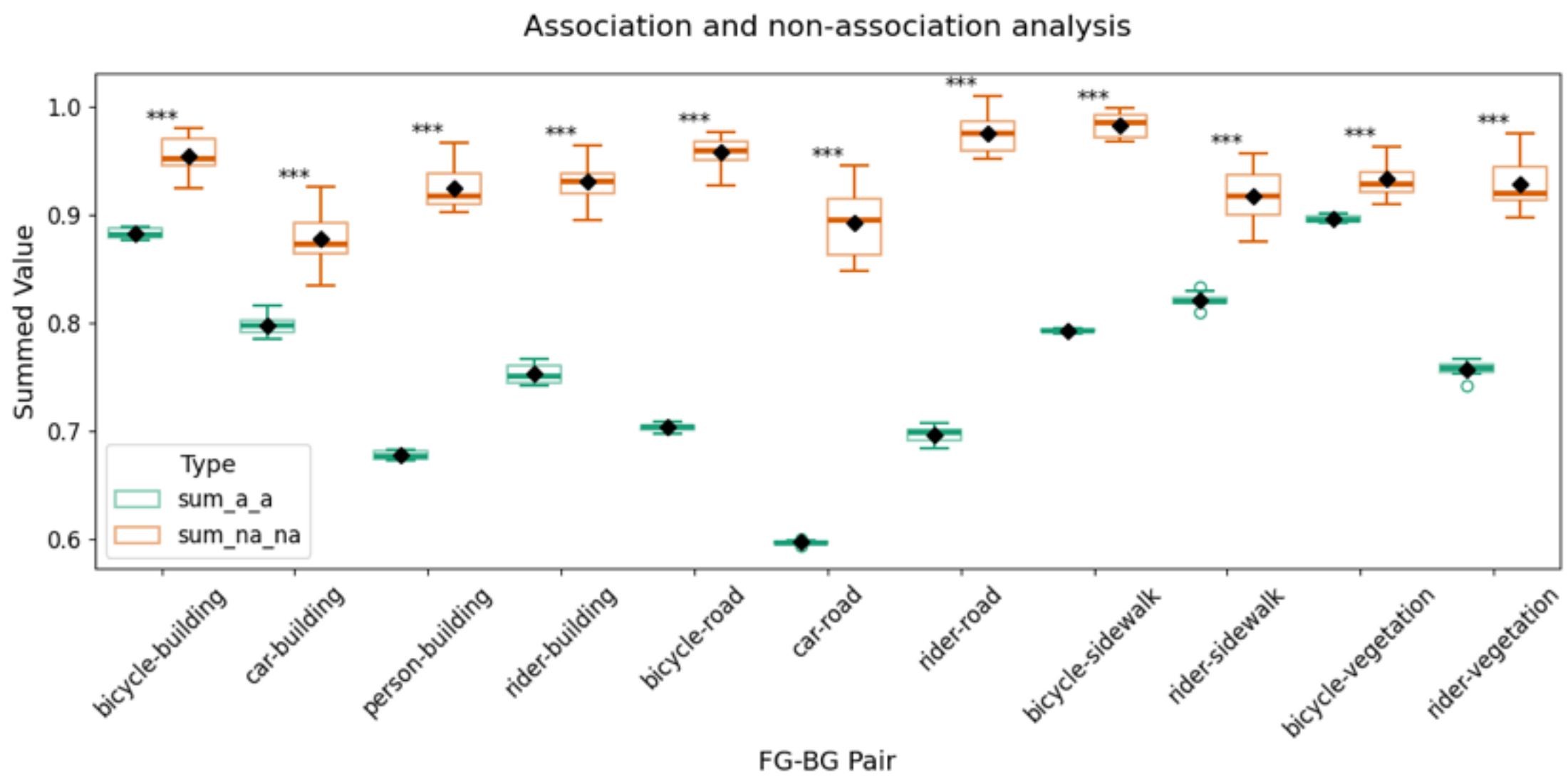

Figure 11: **MMD comparison of associated and non-associated features for the "Res" model on "CST" and "CFV"**. The number of asterisks ("*") indicates the level of statistical significance of a p-value.

#### Future Work

Our results suggest that current DAOD methods may unintentionally reintroduce context bias from the target domain. This highlights a new dimension of the domain adaptation problem and points to the need for bias-aware adaptation strategies that explicitly consider the FG-BG association. Our focus in this work is to show that robust DAOD requires dissociating FG from BG to maintain performance even in divergent domains. The FG–BG association may act as either a spurious or beneficial factor, depending on the stage of the pipeline. During the feature extraction process (primarily handled by the backbone network), entangled features can lead to significant performance drops or complete failure to identify known FG. However, once the FG is identified, BG cues may help determine whether such an association is viable (e.g., identifying a "car" in the "sky" as anomalous).

Since entanglement occurs during feature aggregation, we believe it is due to the pooling mechanism, which aggregates features without regard for object boundaries. To mitigate the entanglement between FG and BG features, pooling mechanisms in DAOD may need to be redesigned. Therefore, building a mask for the object boundary prior to feature aggregation is critical. However, that leads to a chicken-and-egg problem where the feature extraction is required to construct a boundary but the boundary facilitates the object-wise feature extraction. We are actively exploring several potential solutions, including mask-based pooling to emphasize foreground regions, EM-based mask generation, and graph-matching techniques. We believe these approaches can mitigate the identified performance drops, and we plan to extend our study in this direction in future work.

We believe our work opens a novel research direction in DAOD by emphasizing the importance of moving beyond feature alignment to focus on understanding and mitigating causal biases introduced by background context. Future work may explore efficient integration of bias-awareness into end-to-end training pipelines and investigate connections between FG–BG association and broader issues such as spurious correlations and fairness in DAOD.

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

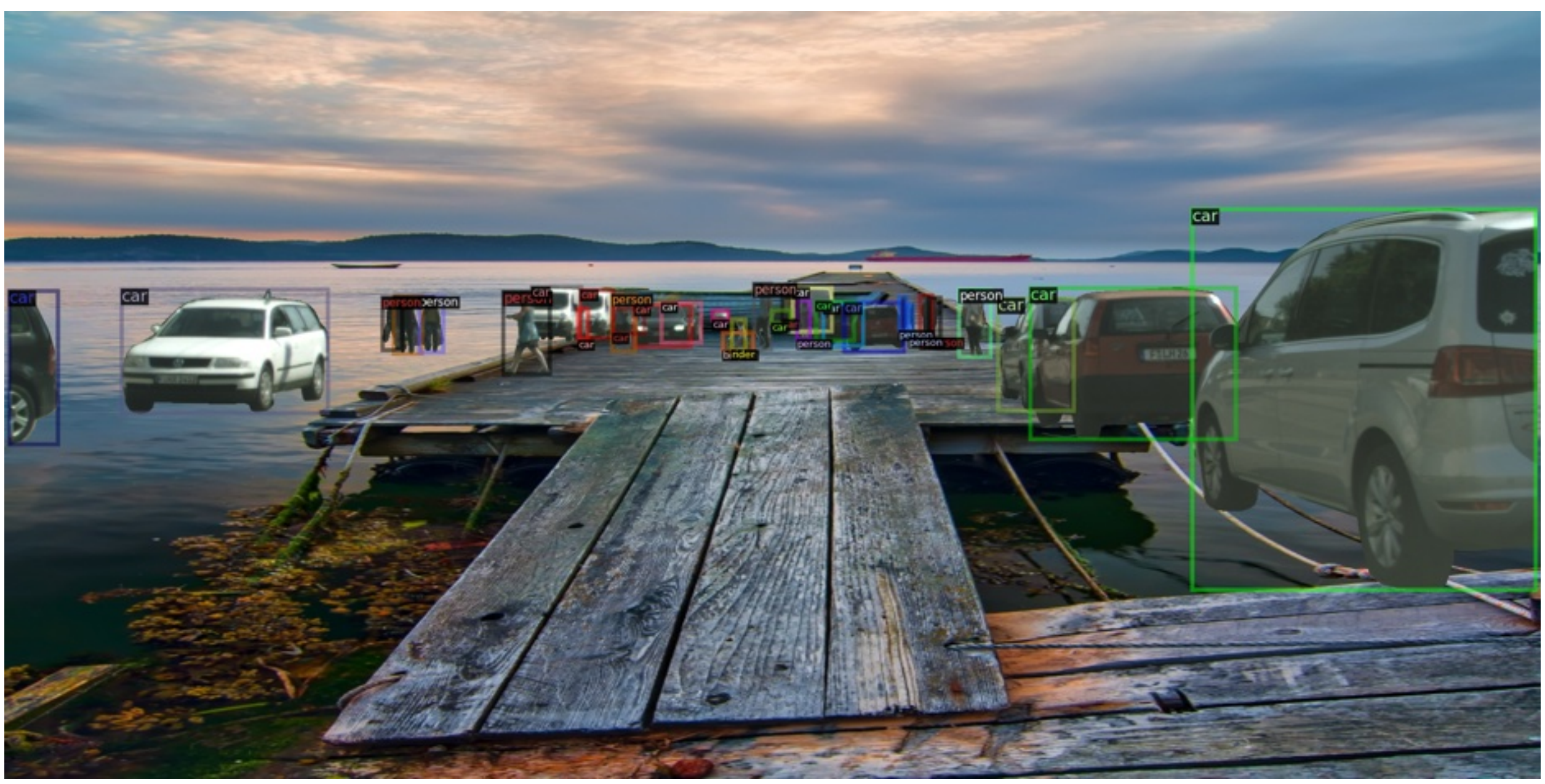

Figure 12: **Example of synthetic image.** We visualize the annotated bounding boxes for foreground objects.

# A Appendix

## A.1 BG-20K synthetic image example

Figure 12 shows an example of superimposing the foreground objects with a random background image from the BG-20K dataset.

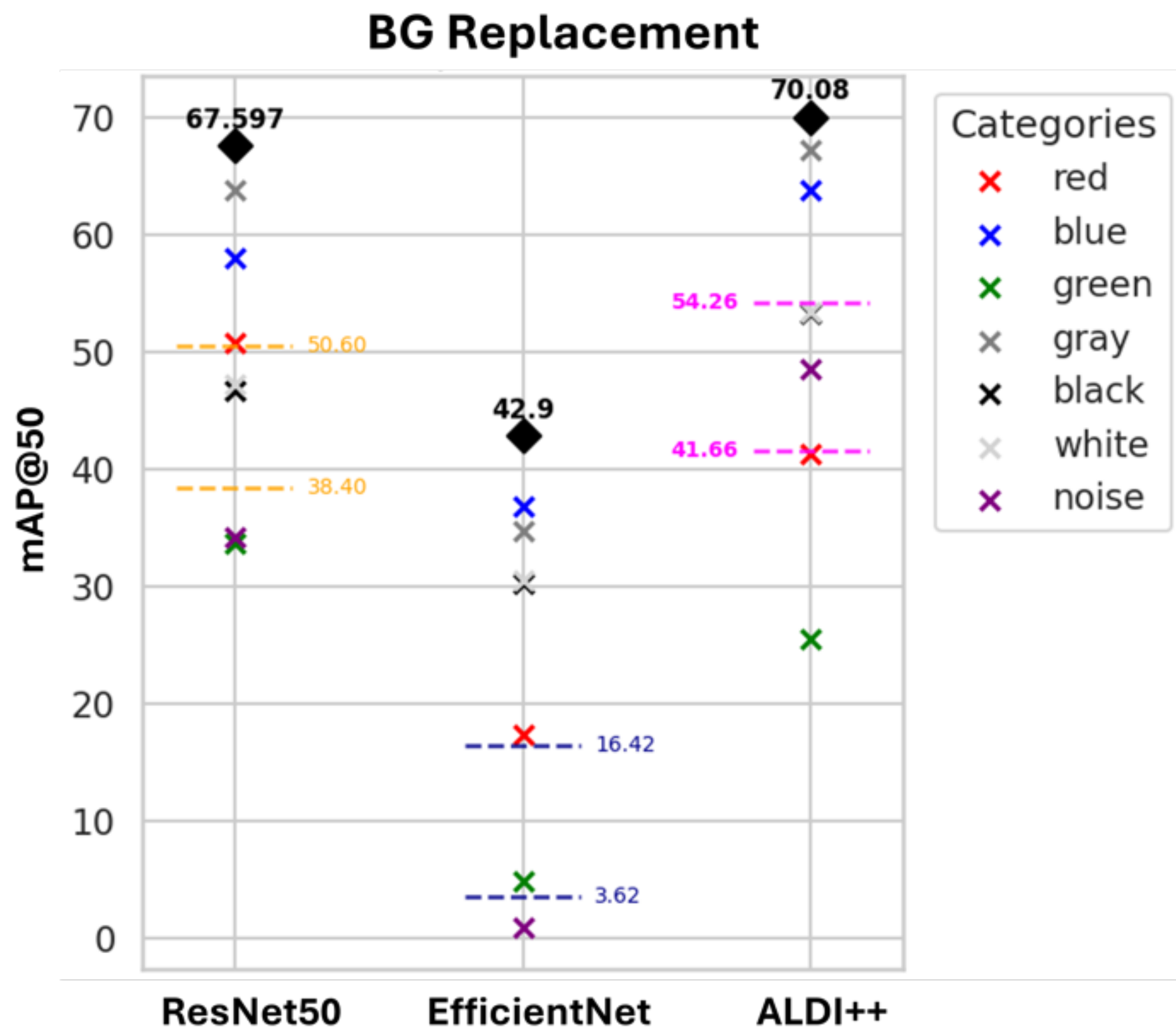

Figure 13: The black diamonds are mAP@50 of each model on the Cityscapes validation set. The different categories ("x") mean types of background (e.g., changed in red). The dashed lines represent the range of random BG experiments we performed. Each number on the figure is mAP@50.

### A.2 Additional BG replacement experiments with simple changes

We conducted fixed BG replacement to ensure a diverse and unbiased distribution of conditions. Specifically, we experimented with different solid background colors and noise-added backgrounds. While we did not perform a detailed statistical analysis, we observed noticeable variations in accuracy across these settings. Our experimental design was therefore structured to progress from simple perturbations to more controlled and feasible analyses, yielding more generalizable and practical results. Figure 13 illustrates the outcome.

### A.3 ViTDeT experiments

Transformer-based detectors can strengthen our experiment, but it is difficult to apply the same process due to their architecture. They reorganize spatial information through a pose-embedding step, which introduces obstacles to tracking spatial information and removes background while capturing foreground-background-related features. So we performed simple experiments using VitDet (Li et al., 2022a). Gradient-based CAM is not suitable; we applied Ablation CAM (Ramaswamy et al., 2020) like method and measured the attention map for each object. For example, we zeroed out for each token for a "car" and computed the weight of the token as the difference Intersection over Union (IoU) between the original bounding box and the new bounding box. If a current target car is not detected after removing the token's activation, the token's weight will be 1.0. We visualize an attention map based on the computed weights and the normalized activation map (see Fig. 14). Because the ViT-based model uses global attention, the map often activates across the entire image. After suppressing the top 20% of the highest responses, the "road" and "buildings" remain prominently highlighted. Besides, we measured the drop rate of mAP@50 by removing

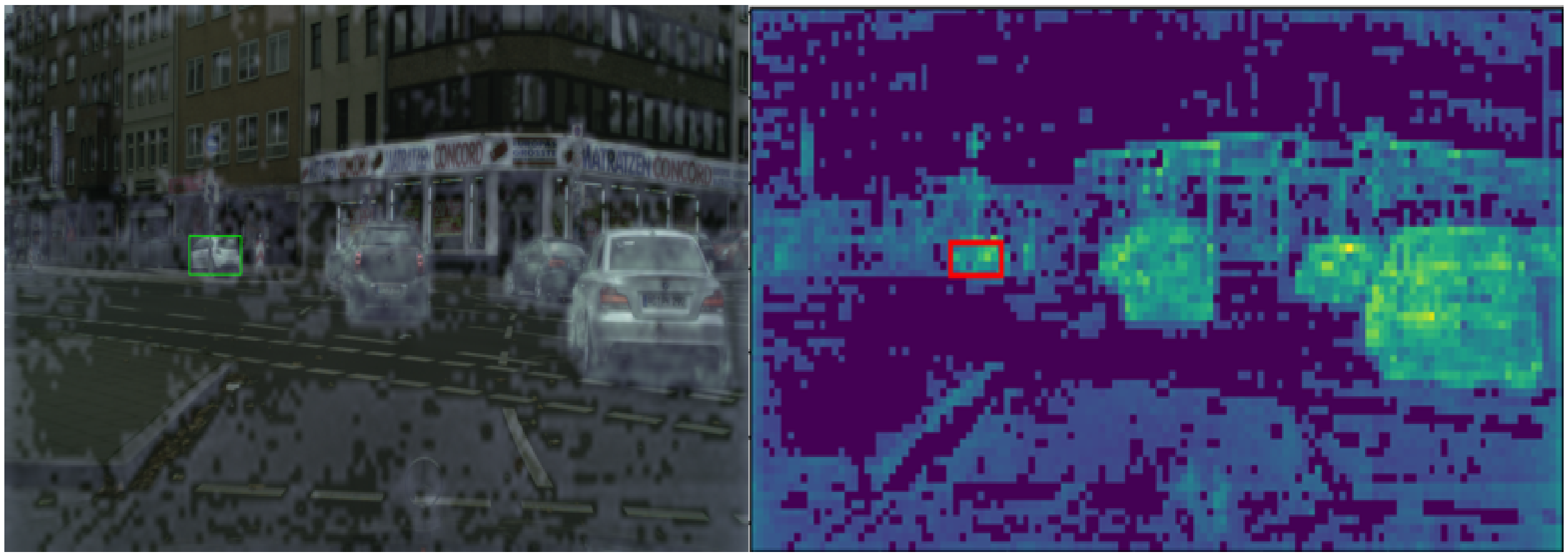

Figure 14: **Left panel**: An attention map (white) is overlaid on the RGB input, with the green box marking the target object. **Right panel**: Attention map with thresholding values. The red bounding box is the same target object.

t

Table 10: Per-background Δ mAP@50 for Cityscapes and Virtual KITTI. mAP@50 on "CSV" is 74.82 and on "VKC" is 79.42. It shows a significant drop even with "CST", which was the training set.

| | road | sidewalk | building | wall | fence | pole | traffic light | traffic sign | vegetation | terrain | sky |
|---|---|---|---|---|---|---|---|---|---|---|---|
| CFV | -16.22 | -15.62 | -9.88 | -1.43 | -1.52 | -4.92 | -0.75 | -0.98 | -7.80 | -3.25 | -0.79 |
| CRV | -14.45 | -11.13 | -10.21 | -0.52 | -0.39 | -3.19 | -0.04 | -0.71 | -5.93 | -1.23 | -0.04 |
| CST | -8.75 | -10.94 | -6.04 | -0.50 | -0.96 | -2.34 | -0.03 | -0.35 | -5.03 | -1.83 | -0.22 |
| CSV | -10.15 | -10.51 | -4.78 | -0.63 | -0.69 | -3.22 | -0.22 | -0.39 | -5.02 | -1.95 | -0.34 |
| KST | -4.77 | -7.01 | -1.01 | -0.95 | -0.52 | -2.96 | +1.58 | +0.11 | -7.07 | -6.29 | -1.29 |

| | terrain | sky | tree | vegetation | building | road | guardrail | traffic sign | traffic light | pole |
|---|---|---|---|---|---|---|---|---|---|---|
| VKC | -1.44 | -0.93 | -1.60 | -0.54 | -0.78 | -0.96 | -0.11 | -0.38 | +0.00 | -0.11 |
| VKF | -10.29 | -7.05 | -12.11 | -8.86 | -4.47 | -7.57 | -5.60 | -2.36 | +0.10 | -0.80 |
| VKM | -1.04 | -1.88 | -3.89 | -0.08 | -1.17 | -1.33 | -0.07 | -0.34 | -0.01 | -0.23 |
| VKR | -3.08 | -0.40 | -2.91 | -3.17 | -2.08 | -2.48 | -0.65 | -1.31 | +0.00 | -0.59 |
| VKO | -0.88 | -1.10 | +1.12 | -1.25 | +0.96 | -3.02 | +0.45 | -0.36 | -0.11 | +0.86 |
| VKS | -2.25 | -1.76 | -3.80 | -0.56 | -1.20 | -1.56 | -0.21 | -0.59 | -0.06 | -0.13 |

the background region in image space. Table 10 demonstrates the drop rate results on Cityscapes and Virtual KITTI datasets. Still, the model expresses significant drops on BG labels. An interesting finding is that the drop rate rarely increases on a few labels, such as "traffic sign" and "traffic light" probably due to the global attention feature. However, this investigation falls beyond our current scope and we aim to analyze transformer-based detectors in future work.

### A.4 MMD with different kernel results

Different from the multiscale kernel used, we utilize RBF kernel to compute MMD. The number of pairs with statistically significant differences differs slightly from the main results, but the findings support the same overall conclusions (see Table 11 and 12).

Table 11: *Gradient* **comparison across different domains with RBF kernel** Each number's denominator is the number of FG–BG association in common across two domains. Cases 1, 2, and 3 are given as C1, C2, and C3. The bolded values represent the dominant case for each dataset-model pair. ∗ indicates that it differs from the result in the main text.

| | **Res** | | | **ALDI++** | | | **Eff** | | |
|---|---|---|---|---|---|---|---|---|---|
| | C1 | C2 | C3 | C1 | C2 | C3 | C1 | C2 | C3 |
| CST - CSV | **2/5**∗ | 1/5 | **2/5**∗ | 4/10 | **6/10** | 0/10 | **3/7** | **3/7** | 1/7 |
| CST - CFV | **11/11** | 0/11 | 0/11 | **15/15** | 0/15 | 0/15 | **8/11**∗ | 2/11 | 1/11∗ |
| CST - CRV | **10/11** | 1/11 | 0/11 | **8/10** | 2/10 | 0/10 | **6/10** | 4/10 | 0/10 |
| CST - KST | **1/1** | 0/1 | 0/1 | **1/1** | 0/1 | 0/1 | **2/3** | 1/3 | 0/3 |
| VKC - VKF | **5/5** | 0/5 | 0/5 | **4/4** | 0/4 | 0/4 | **2/2c** | 0/2 | 0/2 |
| VKC - VKM | **5/7** | 2/7 | 0/7 | **4/4** | 0/4 | 0/4 | 1/5 | **4/5** | 0/5 |
| VKC - VKO | **3/7**∗ | **3/7**∗ | 1/7∗ | **4/4** | 0/4 | 0/4 | 1/5 | **3/5**∗ | 1/5∗ |
| VKC - VKR | **6/6** | 0/6 | 0/6 | **5/5** | 0/5 | 0/5 | 2/5∗ | **3/5**∗ | 0/5 |
| VKC - VKS | **4/6**∗ | 0/6∗ | 1/6∗ | **4/5** | 1/5 | 0/5 | 1/5 | **4/5** | 0/5 |

Table 12: **Associated and non-associated features comparison across different domains with RBF kernel.** Each number's denominator is the number of association in common across two domains. Cases 1, 2, and 3 are given as C1, C2, and C3. "-" is not measurable statistically. The bolded values represent the dominant case for each dataset-model pair. ∗ indicates that it differs from the result in the main text.

| | **Res** | | | **ALDI++** | | | **Eff** | | |
|---|---|---|---|---|---|---|---|---|---|
| | C1 | C2 | C3 | C1 | C2 | C3 | C1 | C2 | C3 |
| CST - CSV | **5/5** | 0/5 | 0/5 | **9/10** | 0/10 | 1/10 | **3/7** | 2/7 | 2/7 |
| CST - CFV | **11/11** | 0/11 | 0/11 | **9/15** | 6/15 | 0/15 | **6/11** | 5/11 | 0/11 |
| CST - CRV | **8/11**∗ | 1/11∗ | 2/11∗ | **7/10** | 3/10 | 0/10 | 2/10 | **6/10** | 2/10 |
| CST - KST | **1/1** | 0/1 | 0/1 | **1/1** | 0/1 | 0/1 | 0/3 | **2/3** | 1/3 |
| VKC - VKF | **3/5**∗ | 1/5 | 1/5∗ | **4/4** | 0/4 | 0/4 | **2/2** | 0/2 | 0/2 |
| VKC - VKM | **4/7** | 2/7 | 1/7 | **4/4** | 0/4 | 0/4 | **3/5** | 1/5 | 1/5 |
| VKC - VKO | **4/7**∗ | 2/7 | 1/7∗ | **4/4** | 0/4 | 0/4 | **4/5** | 1/5 | 1/5 |
| VKC - VKR | **4/6** | 1/6 | 1/6 | **4/5**∗ | 1/5∗ | 0/5 | **4/5** | 1/5 | 0/5 |
| VKC - VKS | **5/6** | 1/6 | 0/6 | **4/5**∗ | 1/5∗ | 0/5 | **3/5** | 1/5 | 1/5 |