# OpenReview forum: "Quantifying Context Bias in Domain Adaptation for Object Detection"
_TMLR — Accepted by TMLR_

### Review · Reviewer_Fnj7 · 2025-08-09

**Summary Of Contributions:**

This paper systematically investigates the role of context bias—specifically foreground–background (FG–BG) association—in cross-domain object detection (DAOD) for the first time in the literature, framing it as a causal relationship and addressing three core research questions (Q1–Q3) through targeted experiments. The authors employ masking, feature-level background manipulation, CAM analysis, and causal calculus to propose the Domain Association Gradient metric, validating their findings across multiple models and datasets.

***Strength***
1. Fills a notable research gap by systematically analyzing FG–BG association in DAOD using causal reasoning, with clear problem formulation (Q1–Q3) and corresponding solutions.
2. Employs a comprehensive experimental design combining masking, CAM, causal analysis, and a novel Domain Association Gradient metric to link FG–BG association strength with cross-domain performance.
3. Demonstrates strong experimental coverage, testing multiple architectures (ResNet-50 FPN, EfficientNet-B0 FPN, YOLOv11, ALDI++) and datasets (Cityscapes, KITTI Semantic, Virtual KITTI, etc.) with diverse performance evaluations.

***Limitation***
1. High computational cost for feature extraction on large datasets, limiting scalability.
2. The novelty of the Domain Association Gradient is underemphasized; differences from existing metrics are not discussed in depth.
3. Lacks empirical evaluation on Transformer-based architectures, despite mentioning their potential to reduce FG–BG dependence.
4. Figures require better readability; e.g., Figure 5 (UMAP) has low color/label distinction, and Figure 10’s red-point annotations are hard to read.

**Audience:**

Yes

**Audience Explanation:**

The findings address a novel causal analysis of context bias in cross-domain object detection, which would be of clear interest to the TMLR audience.

**Broader Impact Concerns:**

There are no apparent broader impact concerns beyond standard considerations for computer vision research, as the work focuses on methodological analysis of context bias in cross-domain object detection without involving sensitive data or high-risk deployment scenarios.

**Claims And Evidence:**

Yes

**Claims Explanation:**

The submission’s main claims are supported by comprehensive experiments, quantitative results, and qualitative analyses that align well with the stated research questions.

**Requested Changes:**

1. Provide more intuitive visualizations (e.g., heatmaps or difference maps) for key results such as drop rate changes and Gradient comparisons.
2. Include main conclusions directly in figure titles/captions to reduce reader inference effort.
3. Proofread for grammar and spelling, especially singular/plural issues (e.g., in Q1 section “These experiments was” should be “were” and on page 13 “divided all datasets into two group” should be “two groups.”}

---

> ### Author Response · Authors · 2025-10-02
> **Rebuttal v1**
>
> We are grateful for your feedback.
>
> 1. We updated our contribution to highlight the novelty of the domain-association gradient. To our knowledge, this is the first work to quantify the phenomenon, so there is no clear precedent for appropriate metrics. Prior domain adaptation research emphasizes foreground feature alignment (e.g., adversarial classifiers, moment matching), which makes existing measures inappropriate to our setting and leaves metric selection ambiguous.
> We have added the following text in Section 1 – “We note that prior domain adaptation research emphasizes FG feature alignment exclusively (e.g., adversarial classifiers and moment matching) and thus our effort is to showcase the gap in the literature.”
>
> 2. Transformer-based detectors can strengthen our experiment, but it is difficult to apply the same process due to their architecture. However, we are interested in the architecture, so we performed simple experiments. Gradient-based CAM does not work, we applied Ablation CAM and measured the attention map for each object. Besides, we measured the drop rate of mAP@50 on the Cityscapes train dataset, which was used for training by removing the background region in image space. There are significant mAP@50 drops, particularly on the shifted domain from a source domain such as Cityscapes foggy Val with road labels. Please check Appendix A.2.
>
> 3. We intentionally used similar colors in Figure 5 for intuitive interpretation. It would be better to clarify that similar colors should be aligned, which indicates it has an analogous distribution, regardless of the types of datasets in the caption. We updated the caption
> We updated Figure 10. Please check.
>
> 4. We tried to convert tables into a merged bar graph but it was confusing because it was too compressed to convey our results with clarity.
>
> 5. We updated the figures and tables’ captions. Updated captions are colorized in red.
>
> 6. It is updated.

---

> > ### Comment · Reviewer_Fnj7 · 2025-11-21
> > **Good modification, but it is suggested to further clarify**
> >
> > Thank you for the revisions and for the clear rebuttal. The paper has improved, and I appreciate the effort. A few issues still need attention.
> >
> > 1. The explanation of the Domain Association Gradient is still limited. Please provide a clearer motivation, compare it with a few possible alternative metrics, and explain why the ratio of drop rate to MMD reflects causal strength. A small example would also help readers understand the idea.
> >
> > 2. The attempt to include Transformer based detectors is appreciated. Although the experiment is limited to a single domain pair, I think it is acceptable as an exploratory addition and does not require further experiments.
> >
> > 3. The writing has improved overall, but several grammatical and formatting issues still remain. I recommend a careful proofreading prior to publication. For example, I only checked the first paragraph of the Introduction:
> >  (a) “there’s” should be “there is”
> >  (b) the citation in 9th line: “Li et al.” is missing a publication year.
> >
> > 4. For the other points I raised in the previous round, I am satisfied with the authors’ responses and the corresponding changes.
> >
> > Overall, the paper is stronger than in the previous version. I encourage the authors to address these final points.

---

> ### Author Response · Authors · 2025-11-25
> **Updated explanation**
>
> Thank you for the comment. We understand that the details were missing and hopefully, we can clear it with the current revision.
>
> 1. For Domain Association gradient, we have added the following explanation in Page 6 of the pdf -
>
> "In the causal graph in Fig 6, we denote FG-BG associations by $A$ and detection performance by $Y$. The causal effect of $A$ on $Y$ like Average Treatment Effect (ATE \citep{naimi2023defining}) can then be given by the interventional contrast as $ ATE = \mathbb{E}\bigl[Y \mid do(A = 1), \ F \bigr]  - \mathbb{E}\bigl[Y \mid do(A = 0),\     F \bigr]$
>
> where $A=1$ corresponds to the presence of the observed FG-BG associations and $A=0$ denotes the intervention that destroys this association. In practice, we estimate this effect using the drop rate (${\Delta{Y}}$), which is the decrease in detection accuracy when we intervene on the background and disrupt FG–BG associations while keeping the foreground object unchanged. Therefore, the drop rate can be seen as an empirical approximation of the average causal effect of $A$ on $Y$.
>
> While the drop rate shows how much performance shifts under an intervention on A, it does not quantify changes in association.
> To disentangle these two factors, we use an MMD-based term that measures the magnitude of the feature distribution shift between associated and non-associated features across inter- and cross-domain contexts. Specifically, the combination of $\Delta{MMD} \approx MMD(\texttt{f2b}) + MMD(\texttt{b2f}) - MMD(\texttt{f2f}) - MMD(\texttt{b2b})$ captures how much foreground ($f$) and background ($b$) features diverge when the association is broken, compared to within-context variation (\texttt{f2f} and \texttt{b2b}). In other words, cross-context feature comparisons, such as \texttt{f2b} and \texttt{b2f}, are expected to yield higher MMD values than within-context comparisons, such as \texttt{f2f} and \texttt{b2b}, because the latter reflect similar contextual structures. However, if FG-BG associations significantly influence object detection, the MMD of cross-context features is smaller than when the associations are absent, yet it remains larger than within-context comparisons. A small value indicates that features capture strong FG-BG associations and a large value implies that features capture weak associations.
> The Domain Association Gradient is then defined as: $\mathrm{Gradient} \approx \frac{\text{change in outcome} \ (\Delta{Y})}{\text{association magnitude} \ (\Delta{MMD})}$
>
> The numerator measures the interventional change in the outcome, while the denominator, $\Delta{MMD}$, quantifies the magnitude of FG-BG associations. Intuitively, if a FG-BG association is strong, the denominator is small and the numerator is large, resulting in high $Gradient$. Conversely, when the association is weak, the $Gradient$ value is low. This makes the $Gradient$ name literal: it’s a rate of change of performance w.r.t. a measure of FG-BG associations."
>
> 3. We apologize for the grammatical mistakes and typos. We have gone through the manuscript thoroughly and hopefully, have tidied the manuscript.

---

### Review · Reviewer_EAoD · 2025-09-03

**Summary Of Contributions:**

The paper contributes towards quantification of the context bias in deep neural network-based classification models. To this end, they consider three questions:
(1) do the classifiers rely upon FG-BG associations during the training process?
(2) does such an association impact FG-BG associations?
(3) does it also play a role in the domain adaptation scenarios  (referred to Domain Adaptation for Object Detection (DAOD))?
They argue, with a series of experiments, that the answer to all three question is yes.

**Additional Comments:**

The authors addressed major concerns during the rebuttal. Namely, there has been a concern about the data imputation, and the authors resolved it by adding extra experiments showing the impact of data imputation. Another concern was to provide the pathways towards the remedies towards addressing the context bias, and the authors provided this as well.

**Audience:**

Yes

**Audience Explanation:**

I think such empirical studies would be interesting to the audience who are interested in training the image classification models, either for practical purposes or for the purposes of scientific research. The approach towards benchmarking is building upon existing techniques, however it gives a useful evidence. The only concern arising would be completeness of such an analysis as discussed in the Claims And Evidence criterion section.

**Broader Impact Concerns:**

No concerns.

**Claims And Evidence:**

Yes

**Claims Explanation:**

I put the answer to No but I would hope that the authors could confirm it during the rebuttal stage. There is one important outstanding question, which is  that I understand that the authors did not justify the missing data imputation procedure. If we black-out the image, as it is done in Figure 4, the risk is that it may happen that the confounder is not the removal of background but the data imputation quality for the removal. Would the statistics change if we replace it with white background? Would they change if the background is random noise? If the background is replaced by simply collating the image as in Figure 7? If, alternatively, the impainting is performed by image generators to allow for the smoother edges around the objects?

Another concern is that the spurious correlations in the input data are reasonably well studied (see, e.g. [1,2] in addition to the cited work), and the questions (1)-(2) are reasonably well researched. What sets this work apart, I think, and it looks like the authors are also suggesting it, is the study in the context of domain adaptation. Nevertheless, a substantial part of the work goes in detail on Questions (1)-(2) while I would think it would be better if the emphasis is put on (3). Therefore, I would think that there should be some link with the theoretical argument or other way to present why the domain generalisation case goes beyond just being a consequence of (1)-(2). In particular, it would be good to show how does that work contrast with the domain generalisation setting [1]?

Also, I think it would be good to add the transformer-based architectures, as it might be of particular interest to see if the conclusions hold across the architectures.

Finally, I think it would really strengthen the paper if the authors could outline the possible remedies for the proposed performance drop resulting from that analysis.

[1] Qin et al (2024) Revisiting Spurious Correlation in Domain Generalization, Arxiv
[2] LaBonte et al (2024) The Group Robustness is in the Details: Revisiting Finetuning under Spurious Correlations

**Requested Changes:**

As described above, clarifying on data imputation is a crucial concern.

Besides, the rest of the discussions on the claims and evidence must be addressed.

---

> ### Author Response · Authors · 2025-10-02
> **Rebuttal v1**
>
> We sincerely appreciate your comments. We have carefully revised the manuscript to address each point raised.
>
> 1. This is an important question and we thank the reviewer for asking for clarification. In order to move away from the data imputation quality problem, we chose random background images to overlay the foreground objects over 6 different runs. The results are given in Table 4. But we realized we have not performed the experiment using a solid color background or random noise. Thus, in Appendix A.1, we provide the detection results using those backgrounds. We can see that different BGs affect the detection accuracy differently and they are not the same across the models. We submit this as an example of the FG-BG association using simple BG substitutions.
>
> 2. Thank you for pointing us to additional references. We intended to build a solid theoretical and empirical foundation for addressing Question (3). In contrast to [1], which applies propensity score weighting (via FFT and K-clustering) to study spurious features in classification, our focus is on domain-adaptive object detection (DAOD). This setting is inherently more complex, since it involves multiple objects per image, diverse and salient backgrounds, and structured FG–BG associations that go beyond the simpler classification case.
> Our core idea is to investigate how FG–BG associations act as confounders in DAOD. We do not treat background as purely spurious; rather, we highlight that it can play different roles, including in post-processing or feature alignment. From a domain-generalization perspective, we show that the domain shift between BG–BG pairs is substantially larger than between FG–FG pairs. This suggests that achieving robust generalization requires explicitly addressing the FG–BG association, rather than viewing it as spurious correlations. We see this as a key point of departure from (1)–(2), and believe it frames an important direction for future work: how to effectively align or utilize FG–BG associations across domains while retaining background cues that remain useful in real-world applications. For example, other FG objects can be BG to other FG. Contextually meaningful BG with FG can be helpful to predict edge cases of FG.
> [2] compares different types of methods for alleviating spurious features on classification tasks with different domain datasets. It is not exactly related work but it can provide insight into mitigation methods for DAOD.
>
> 3. Transformer-based detectors can strengthen our experiment, but it is difficult to apply the same process due to their architecture. However, we are interested in the architecture, so we performed simple experiments. Gradient-based CAM does not work; we applied Ablation CAM and measured the attention map for each object. Besides, we measured the drop rate of mAP@50 on the Cityscapes train dataset, which was used for training by removing the background region in image space. There are significant mAP@50 drops, particularly on the shifted domain from a source domain such as Cityscapes foggy Val with road labels. Please check Appendix A.2.
>
> 4. As we have mentioned in the Future Work section (Section 5), we are actively exploring several potential remedies, including mask-based pooling to emphasize foreground regions, EM-based mask generation, and graph-matching techniques. We believe these approaches can mitigate the identified performance drops, and we plan to extend our study in this direction in future work.

---

> ### Comment · Reviewer_EAoD · 2025-10-06
>
> First, I would like to thank the authors for preparing the revision and the rebuttal.
> 1. Thank you, that helps answer the question, let me read the updated version and then correct the score
> 2. It seems to me that the essence of the response is in agreement in the comment, which is that the main contribution is in Q3. The current revision of the text help emphasise it. " We see this as a key point of departure from (1)–(2), and believe it frames an important direction for future work: how to effectively align or utilize FG–BG associations across domains while retaining background cues that remain useful in real-world applications." That particular point needs clarification: while it is good that the authors think of it in the context of the future work, it would be useful to know in which way this work helps answer this question.
>
> "" We intended to build a solid theoretical and empirical foundation for addressing Question (3). In contrast to [1], which applies propensity score weighting (via FFT and K-clustering) to study spurious features in classification, our focus is on domain-adaptive object detection (DAOD). " Is there going to be another revision, or how would it fit into this paper?
>
> 3. Thank you, I will check appendix A.2
> 5. On the mitigation side, I would suggest to expand on the takeaway message (perhaps in section 5 Discussion and Conclusion), which would incorporate the details of explored remedies.
>
> In the meantime, reading the updated draft and will update the score in due course.

---

> > ### Author Response · Authors · 2025-10-06
> >
> > Thank you for the additional feedback.
> >
> > 2. Our focus through this work is to show that robust DAOD requires dissociating FG from BG in order to maintain the performance level even in the divergent domains. The FG–BG association may act as a spurious or beneficial factor, depending on the stage of the pipeline. During the feature extraction process (done mostly by the backbone network), if the features get entangled, that can lead to large performance drops or complete non-identification of known FG. However, once the FG has been identified, it helps to provide BG cues to understand whether such an association is viable or not. For example, a car being in sky can account for such an example where we want to identify the car and then flag the association between the car and sky as unusual.
> >
> > About [1], we try to showcase their method and how ours differs in methodology. We want to state that their application is classification while ours is DAOD. Therefore, while their method is interesting, it actually is not comparable. We don't think using FFT in image space is helpful but it can be an interesting topic of tensor decomposition for DAOD. K-clustering in feature space is one of the options for future work to make it separate FG and BG features. Propensity-score ideas are intriguing, but applying them to detectors is non-trivial because “foreground” in one class can act as “background” for another. This complexity likely explains why the prior propensity-score approach underperforms on more complex domains (e.g., RotatedMNIST, VLCS, TerraIncognita) compared with simpler setups (e.g., ColoredMNIST, PACS, OfficeHome).
> >
> > We updated the Future Work section in red color
> >
> > 4. Since the entanglement occurs during the feature aggregation process, we believe it is due to the pooling mechanism which aggregates features without any thought of object boundaries. To mitigate the entanglement of FG and BG features, we believe pooling steps may be redesigned for DAOD. Therefore, building a mask for object boundary is crucial before the feature extraction process. However, that leads to a chicken-and-egg problem which we would have to solve in the process. As we have mentioned in the Future Work section (Section 5), we are actively exploring several potential remedies, including mask-based pooling to emphasize foreground regions, EM-based mask generation, and graph-matching techniques. We believe these approaches can mitigate the identified performance drops, and we plan to extend our study in this direction in future work.
> >
> > We updated the Future Work section in red color

---

### Review · Reviewer_46Cs · 2025-09-23

**Summary Of Contributions:**

The paper empirically probes foreground–background (FG–BG) context bias in domain-adaptive object detection (DAOD). It conducts three complementary studies: (Q1) a feature-wise background removal test to quantify detection drop rates per FG–BG pair; (Q2) a CAM-based causal stress test (Smooth-GradCAM++) that progressively masks CAM-activated background to measure causal contributions; and (Q3) a Domain Association Gradient that aggregates drop-rate and MMD distances across associated vs non-associated feature sets to assess how source-learned context ties carry into the target domain. Results suggest DAOD models indeed internalize spurious FG–BG associations that persist under shift and impact performance.

Strength:

1. The triangulation via (Q1) feature-wise removal, (Q2) CAM-threshold “intervention,” and (Q3) MMD-based gradient offers convergent evidence rather than relying on a single proxy.

2. The drops are computed on true positives, repeated six times, with a Wilcoxon signed-rank test acknowledging non-normality.

3. Algorithms for feature removal (Alg. 2), CAM causal test (Alg. 3), and feature extraction (Alg. 4) are explicit—good for reproducibility.

Weakness:

1.	Causal identification remains a heuristic. CAM-based masking is not a principled intervention on the data-generating process; CAMs are class-score–dependent and noisy. The “causal” conclusions are suggestive trends, not identifiability guarantees. Stronger sanity checks or counterfactual controls would help.

2.	Definition of association is model-contingent. Labeling features as associated/non-associated hinges on detector failure after BG suppression, potentially conflating context reliance with general signal degradation (e.g., early-layer masking alters low-level cues).

3.	The Domain Association Gradient mixes drop rates with sums/differences of several MMDs (f2f, b2b, f2b, b2f) without robustness analysis to kernel type, bandwidth, or feature normalization. Conclusions could be sensitive to these choices.

4.	Feature-wise background removal test intervenes at specific shallow blocks (e.g., res2.2); context representations can reside deeper. Without a layer sweep or backbone-homologous mapping, estimates of reliance may be biased.

5.	The study focuses on a few architectures and mAP@50. Including transformer-based detectors and mAP@[.5:.95] could strengthen generality.

6.	There are many FG–BG pairs, layers, thresholds, and domains. Beyond per-pair Wilcoxon tests, there’s no global multiplicity control or effect-size reporting.

**Audience:**

Yes

**Audience Explanation:**

The paper provides a useful and carefully constructed diagnostic suite for context bias in DAOD, with multiple complementary probes and clear procedures.

**Claims And Evidence:**

Yes

**Claims Explanation:**

The work has a strong empirical framing with a causal interpretation in heuristic.

**Requested Changes:**

1. Can the authors add sanity checks: (a) random masks with matched area; (b) class-agnostic saliency controls; (c) counterfactual overlays (BG replacement) to verify specificity beyond CAM artifacts?

2. Quantify false positives/negatives when labeling instances as associated via Algorithm 4. For example, measure how often association flips with small changes in masking strength.

3. The authors can add kernels, bandwidth selection, and feature normalization to provide a robust MMD sensitivity.

4. It would also be worthwhile to include results for interventions at multiple backbone depths (or principled mapping across backbones) to establish the stability of Q1 findings.

5. Add deformable DETR/ViT-based detectors to test metric/model dependence.

6. Given the diagnosis, it would be interesting to show at least one practical mitigation (e.g., background randomization, cut–paste with domain randomization) and quantify the reduction in gradient and mAP impact.

---

> ### Author Response · Authors · 2025-10-02
> **Rebuttal v1**
>
> Thank you for your feedback. Below we address each comment point-by-point.
> 1. The question is a bit ambiguous to us – for (a) do we apply inaccurate masks to see their effect? For (b), we have used the CAM to get the area where activation occurs after applying the masks. To assess CAM artifacts, we measured the CAM hit ratio. Because CAMs can be unreliable, especially in multi-object scenes, we computed the ratio using panoptic segmentation masks for each true-positive object. We omit the full results for brevity. Table 6 reports a representative subset. The strongest CAM responses occur in the target object’s foreground, and Figure 8 shows the bin-thresholded visualization.
> For (c),  we performed BG replacement for the sanity check. Please check updated Appendix A.1.
>
> 2. Applying the same algorithm to false positives/negatives (FP/FN) is ambiguous because it is unclear whether these errors arise from foreground–background associations or from the model training itself. Therefore, we excluded FP/FN cases from our experiments and focused our analysis on true positives.
>
> 3. We include Domain Association Gradients and association comparison results using an RBF kernel. The number of pairs with statistically significant differences differs slightly from our previous results, but the findings support the same overall conclusions. Please check Appendix A3.
>
> 4. We considered that approach, but it would eliminate too much relational information, including foreground related features. The FG–BG association reflects high-level features derived from low-level cues. Accordingly, we disrupt background low-level features, which in turn breaks the FG–BG association.
>
> 5. Transformer-based detectors can strengthen our experiment, but it is difficult to apply same process due to its architecture. However, we are interested in the architecture, we performed simple experiments. Gradient-based CAM does not work, we applied Ablation CAM and measured the attention map for each object. Besides, we measured the drop rate of mAP@50 on Cityscapes train dataset which used for training by removing background region in image space. There are significant mAP@50 drops, particularly on the shifted domain from a source domain such as cityscapes foggy Val with road labels. Please check Appendix A.2
>
> 6. As we have mentioned in the Future work section (Section 5), we are actively exploring several potential remedies, including mask-based pooling to emphasize foreground regions, EM-based mask generation, and graph-matching techniques. We believe these approaches can mitigate the identified performance drops, and we plan to extend our study in this direction in future work.

---

### Decision · Action_Editor_RVMm · 2025-12-02

**Recommendation:** Accept with minor revision

**Additional Comments:**

As noted by Reviewer Fnj7, several grammatical and formatting issues remain and should be corrected in the final version. For example,

- Figures and tables should be positioned at the top of the page to improve readability and consistency. For reference, see examples such as: https://openreview.net/forum?id=2cxxZI2LOL

- The phrase “it’s” (below Equation 4) should be corrected to “it is.” More generally, the manuscript would benefit from a thorough proofreading to address remaining grammatical and typographical issues prior to publication.

**Audience:**

Yes

**Audience Explanation:**

The empirical analysis and findings address an important issue in object detection—namely, the influence of context bias—which is of clear relevance to the TMLR audience, particularly those working in computer vision and applied machine learning.

**Claims And Evidence:**

Yes

**Claims Explanation:**

This paper presents a systematic investigation of foreground–background (FG–BG) context bias in domain-adaptive object detection (DAOD). Using feature-wise background removal, CAM-based causal analysis, and a newly introduced Domain Association Gradient metric, the authors demonstrate—across multiple architectures and datasets—that DAOD models tend to internalize spurious FG–BG associations that persist under domain shifts and substantially affect performance.

All three reviewers recognize several key strengths: (1) the systematic and comprehensive scope of the study; (2) strong empirical validation; and (3) clear methodology with good reproducibility. However, they also raised multiple initial concerns, including: (1) the heuristic nature of the causal identification strategy; (2) questions about the design of the Domain Association Gradient and the data imputation procedure; (3) limited evaluation on alternative architectures such as Transformer-based detectors; and (4) the absence of practical mitigation strategies. The authors’ rebuttal satisfactorily addressed most of the major concerns. Two reviewers ultimately provided “leaning accept” recommendations, and the third expressed continued support following the discussion. Collectively, the reviewers agree that the manuscript has improved and that the primary concerns have been addressed.

The AE concurs with the reviewers’ assessment and recommends acceptance pending minor revision. The authors are encouraged to incorporate relevant clarifications raised during the discussion.